# Soft Diffusion
# Score Matching For General Corruptions

## Abstract

We define a broader family of corruption processes that generalizes previously known diffusion models. To reverse these general diffusions, we propose a new objective called Soft Score Matching that provably learns the score function for any linear corruption process and yields state of the art results for CelebA. Soft Score Matching incorporates the degradation process in the network. Our new loss trains the model to predict a clean image, *that after corruption*, matches the diffused observation. We show that our objective learns the gradient of the likelihood under suitable regularity conditions for a family of corruption processes. We further develop a principled way to select the corruption levels for general diffusion processes and a novel sampling method that we call Momentum Sampler. We show experimentally that our framework works for general linear corruption processes, such as Gaussian blur and masking. We achieve state-of-the-art FID score 1.85 on CelebA-64, outperforming all previous linear diffusion models. We also show significant computational benefits compared to vanilla denoising diffusion.

## 1 Introduction

Score-based models (Song & Ermon, 2019; 2020; Song et al., 2021b) and Denoising Diffusion Probabilistic Models (DDPMs) (Sohl-Dickstein et al., 2015; Ho et al., 2020; Song et al., 2021a) are two powerful classes of generative models that produce samples by inverting a diffusion process. These two classes have been unified under a single framework (Song et al., 2021b) and are widely known as diffusion models. Diffusion modeling has found great success in a wide range of applications (Croitoru et al., 2022; Yang et al., 2022), including image (Saharia et al., 2022a; Ramesh et al., 2022; Rombach et al., 2022; Dhariwal & Nichol, 2021), audio (Kong et al., 2021; Richter et al., 2022; Serrà et al., 2022), video generation (Ho et al., 2022b), as well as solving inverse problems (Daras et al., 2022; Kadkhodaie & Simoncelli, 2021; Kawar et al., 2022; 2021; Jalal et al., 2021; Saharia et al., 2022b; Laumont et al., 2022; Whang et al., 2022; Chung et al., 2022).

Karras et al. (2022) analyze the design space of diffusion models. The authors identify three stages: i) the noise scheduling, ii) the network parametrization (each one leads to a different loss function), iii) the sampling algorithm. We argue that there is one more important step: *choosing how to corrupt*. Typically, the diffusion is additive noise of different magnitudes (and sometimes input rescalings). There have been a few recent attempts to use different corruptions (Deasy et al., 2021; Hoogeboom et al., 2022a;b; Avrahami et al., 2022; Nachmani et al., 2021; Johnson et al., 2021; Lee et al., 2022; Ye et al., 2022), but the results are usually inferior to diffusion with additive noise. Also, a common framework on how to properly design general corruption processes is missing.

We present such a principled framework for learning to invert a general class of corruption processes. We propose a new objective called *Soft Score Matching* that provably learns the score for any regular linear corruption process. Soft Score Matching incorporates the filtering process in the network and trains the model *to predict a clean image that after corruption matches the diffused observation*.

Our theoretical results show that Soft Score Matching learns the score (i.e. likelihood gradients) for corruption processes that satisfy a regularity condition that we identify: the diffusion must transform any image into any other image with nonzero likelihood. Using our method and Gaussian Blur paired with little noise as the diffusion mechanism, we achieve state-of-the-art FID on CelebA (FID 1.85) for linear diffusion models. We also show that our corruption process leads to generative models that are faster compared to vanilla Gaussian denoising diffusion.

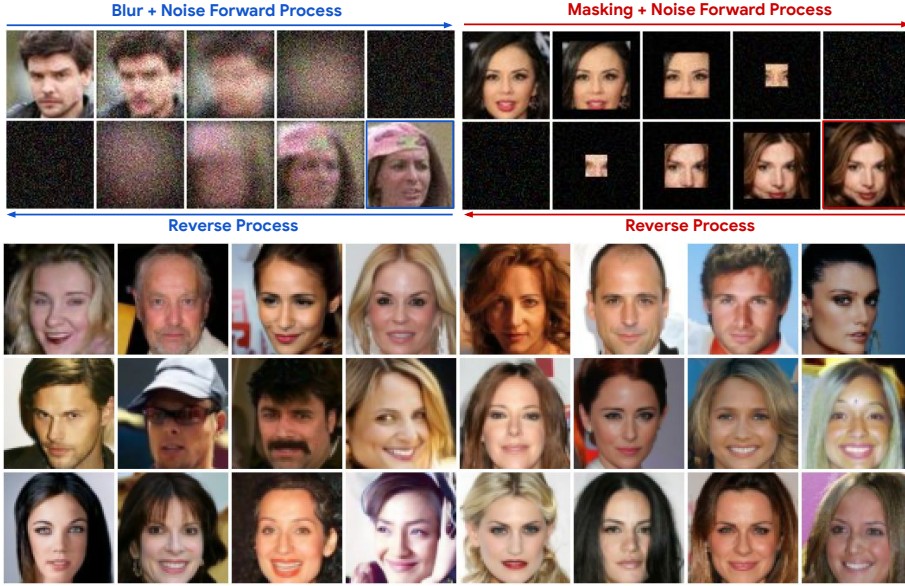

Figure 1: Top two rows: Demonstration of our generalized diffusion method. Instead of corrupting by only adding noise, we propose a framework to provably learn the score function to reverse any linear diffusion (left: blur and noise, right: masking and noise). Our (blur and noise) models achieve state-of-the-art FID score $\mathbf{1.85}$ for linear diffusion models on CelebA-64. Uncurated samples shown in the last three rows.

**Our contributions:** a) We propose a learning objective that: i) provably learns the score for a wide family of regular diffusion processes and ii) enables learning under limited randomness in the diffusion. b) We present a principled way to select the intermediate distributions. Our method minimizes the Wasserstein distance along the path from the initial to the final distribution. c) We propose a novel sampling method that we call Momentum Sampler: our sampler uses a convex combination of corruptions at different diffusion levels and is inspired by momentum methods in optimization. d) We train models on CelebA and CIFAR-10. Our trained models on CelebA achieve a new state-of-the-art FID score of $1.85$ for linear diffusion models while being significantly faster compared to models trained with vanilla Gaussian denoising diffusion.

## 2 BACKGROUND

Diffusion models are generative models that produce samples by inverting a corruption process. The corruption level is typically indexed by a time $t$, with $t = 0$ corresponding to clean and $t = 1$ to fully corrupted images. The diffusion process can be discrete or continuous. The two general classes of diffusion models are Score-Based Models (Song & Ermon, 2019; 2020; Song et al., 2021b) and Denoising Diffusion Probabilistic Models (DDPMs) (Sohl-Dickstein et al., 2015; Ho et al., 2020).

The typical diffusion in score-based modeling is additive noise of increasing magnitude. The perturbation kernel at time $t$ is: $q_t(\boldsymbol{x}_t|\boldsymbol{x}_0) = \mathcal{N}(\boldsymbol{x}_t|\boldsymbol{\mu} = \boldsymbol{x}_0, \boldsymbol{\Sigma} = \sigma_t^2 \boldsymbol{I})$, where $\boldsymbol{x}_0 \sim q_0$ is a clean image. Score models are trained with the Denoising Score Matching (DSM) objective:

$$\min_{\theta} \mathbb{E}_{t \sim U[0,1]} w_t \left[ \mathbb{E}_{(\boldsymbol{x}_0, \boldsymbol{x}_t) \sim q_0(\boldsymbol{x}_0) q_t(\boldsymbol{x}_t|\boldsymbol{x}_0)} ||s_\theta(\boldsymbol{x}_t|t) - \nabla_{\boldsymbol{x}_t} \log q_t(\boldsymbol{x}_t|\boldsymbol{x}_0)||^2 \right], \tag{1}$$

where $w_t$ scales differently the weights of the inner objectives. If we train for each noise level $t$ independently, given enough data and model capacity, the network is guaranteed to recover the gradient of the log likelihood (Vincent, 2011), known as the score function. In other words, the model $s_\theta(\boldsymbol{x}_t|t)$ is trained such that: $s_\theta(\boldsymbol{x}_t|t) \approx \nabla_{\boldsymbol{x}_t} \log q_t(\boldsymbol{x}_t)$. In practice, we use parameter sharing and conditioning on time $t$ to learn all the scores. Once the model is trained, we start from a sample of the final distribution, $q_1$, and then use the learned score to gradually denoise it (Song & Ermon, 2019; 2020). The final variance $\sigma_1^2$ is selected to be very large such that the distribution $q_1$ is approximately Gaussian, i.e. the signal to noise ratio tends to $0$.

DDPMs corrupt by rescaling the input images and by adding noise. The corruption can be modelled with a Markov chain with perturbation kernel $q_t(\boldsymbol{x}_t|\boldsymbol{x}_{t-\Delta t}) = \mathcal{N}(\boldsymbol{x}_t|\boldsymbol{\mu} = \sqrt{1 - \beta_t}\boldsymbol{x}_{t-\Delta t}, \boldsymbol{\Sigma} =$

$\beta_t \boldsymbol{I}$). Typically, $\beta_1 = 1$ and hence $q_1 = \mathcal{N}(0, I)$. DDPMs are also trained with the DSM objective which is derived by minimizing an evidence lower bound (ELBO) (Ho et al., 2020).

In their seminal work, Song et al. (2021b) observe that the diffusions of both Score-Based models and DDPMs can be expressed as solutions of Stochastic Differential Equations (SDEs) of the form:

$$\mathrm{d}\boldsymbol{x} = \boldsymbol{f}(\boldsymbol{x}, t)\mathrm{d}t + g(t)\mathrm{d}\boldsymbol{w}, \tag{2}$$

where $\boldsymbol{w}$ is the standard Wiener process. Particularly, Score-Based models use: $f(\boldsymbol{x}, t) = \boldsymbol{0}, g(t) = \sqrt{\frac{\mathrm{d}\sigma_t^2}{\mathrm{d}t}}$ and DDPMs use: $\boldsymbol{f}(\boldsymbol{x}, t) = -\frac{1}{2}\beta_t\boldsymbol{x}$, $g(t) = \sqrt{\beta_t}$. As explained earlier, for Score-Based models we need large noise at the end for the final distribution to approximate a normal distribution. Hence, the corresponding SDE is named **V**ariance **E**xploding (VE) SDE (Song et al., 2021b). DDPMs usually have a final distribution of unit variance and hence their SDE is known as the **V**ariance **P**reserving (VP) SDE (Song et al., 2021b). Song et al. (2021b) also propose another SDE with bounded variance, the subVP-SDE, that experimentally yields better likelihoods.

For both Score-Based models and DDPMs, Eq. (2) is known as the Forward SDE. This SDE is reversible Anderson (1982) and the Reverse SDE is given below:

$$\mathrm{d}\boldsymbol{x} = \left[\boldsymbol{f}(\boldsymbol{x}, t) - g^2(t)\nabla_{\boldsymbol{x}}\log q_t(\boldsymbol{x})\right]\mathrm{d}t + g(t)\mathrm{d}\bar{\boldsymbol{w}}, \tag{3}$$

where $\bar{\boldsymbol{w}}$ is the reverse time standard Wiener process. Typically, $\nabla_{\boldsymbol{x}}\log q_t(\boldsymbol{x})$ is approximated by $s_\theta(\boldsymbol{x}_t|t)$ and samples are generated by solving the Reverse SDE (Song et al., 2021b).

## 3 METHOD

Our framework for training diffusion models with more general corruptions includes three components: i) the training objective, ii) the sampling, iii) the scheduling of the corruption mechanism.

### 3.1 TRAINING OBJECTIVE

We study corruption processes of the form:

$$\boldsymbol{x}_t = \boldsymbol{C}_t\boldsymbol{x}_0 + s_t\boldsymbol{\eta}_t, \tag{4}$$

where $\boldsymbol{C}_t : \mathbb{R}^n \to \mathbb{R}^n$ is a deterministic linear operator, $\boldsymbol{\eta}_t$ is a Wiener process, and $s_t$ is a non-negative scalar controlling the noise level at time $t$, and $\boldsymbol{x}_0 \sim q_0(\boldsymbol{x})$. We further denote with $\sigma_t^2$ the variance of the noise at level $t$ and we assume that it is a non-decreasing function of $t$. Unless stated otherwise, we assume that time is continuous and runs from $t = 0$ to $t = 1$. Additionally, we assume that at $t = 0$, we have $\boldsymbol{C}_0 = \boldsymbol{I}_{n \times n}$ and $\sigma_0 = 0$, i.e. $t = 0$ corresponds to natural images. We also assume that recovering $\boldsymbol{x}_0$ from $\boldsymbol{x}_t$ is harder as $t$ gets larger (i.e., entropy of $q_t(\boldsymbol{x}_0 \mid \boldsymbol{x}_t)$ increases with $t$). Eq. (4) defines a general class of diffusion processes, that includes (as special cases) the VE, VP and subVP SDEs used in Song et al. (2021b). Our diffusion is the sum of a deterministic linear corruption of $\boldsymbol{x}_0$ and a stochastic part that progressively adds noise. For any corruption process of this family, we are interested in learning the scores, i.e. $\nabla_{\boldsymbol{x}_t}\log q_t(\boldsymbol{x}_t)$ for all $t$.

For the vanilla Gaussian denoising diffusion, the celebrated result of Vincent (2011) shows that we only need access to the gradient of the conditional log-likelihood, $\nabla_{\boldsymbol{x}_t}\log q_t(\boldsymbol{x}_t|\boldsymbol{x}_0)$, in order to learn the score, $\nabla_{\boldsymbol{x}_t}\log q_t(\boldsymbol{x}_t)$. By revisiting the proof of Vincent (2011), we find that this is actually true for a wide set of corruption processes, as long as some mild technical conditions are satisfied. In fact, the following general Theorem holds:

**Theorem 3.1.** *Let $q_0, q_t$ be two distributions in $\mathbb{R}^n$. Assume that all conditional distributions, $q_t(\boldsymbol{x}_t|\boldsymbol{x}_0)$, are fully supported and differentiable in $\mathbb{R}^n$. Let:*

$$J_1(\theta) = \frac{1}{2}\mathbb{E}_{\boldsymbol{x}_t \sim q_t}\left[||\boldsymbol{s}_\theta(\boldsymbol{x}_t) - \nabla_{\boldsymbol{x}_t}\log q_t(\boldsymbol{x}_t)||^2\right], \tag{5}$$

$$J_2(\theta) = \frac{1}{2}\mathbb{E}_{(\boldsymbol{x}_0, \boldsymbol{x}_t) \sim q_0(\boldsymbol{x}_0)q_t(\boldsymbol{x}_t|\boldsymbol{x}_0)}\left[||\boldsymbol{s}_\theta(\boldsymbol{x}_t) - \nabla_{\boldsymbol{x}_t}\log q_t(\boldsymbol{x}_t|\boldsymbol{x}_0)||^2\right]. \tag{6}$$

*Then, there is a universal constant $C$ (that does not depend on $\theta$) such that: $J_1(\theta) = J_2(\theta) + C$.*

Theorem 3.1 implies that minimizing the second function is equivalent to minimizing the first one. The second function is nothing else than the DSM objective. Our main observation is that noise is not always necessary for learning the score using the DSM objective. A necessary condition is that the corruption process gives non-zero probability to all $\boldsymbol{x}_t$ for any image $\boldsymbol{x}_0$. This is easily achieved by adding Gaussian noise, but this is not the only option. The proof of this Theorem is deferred in the Appendix and it is following the calculations of Vincent (2011).

**Network parametrization.** For the class of diffusion processes given by Eq. (4), we have that: $q_t(\boldsymbol{x}_t|\boldsymbol{x}_0) = \mathcal{N}\left(\boldsymbol{x}_t; \boldsymbol{\mu} = \boldsymbol{C}_t\boldsymbol{x}_0, \boldsymbol{\Sigma} = \sigma_t^2\boldsymbol{I}\right)$ and hence the objective becomes:

$$L(t) = \frac{1}{2}\mathbb{E}_{(\boldsymbol{x}_0,\boldsymbol{x}_t)\sim q_0(\boldsymbol{x}_0)q_t(\boldsymbol{x}_t|\boldsymbol{x}_0)}\left[\left|\left|\boldsymbol{s}_\theta(\boldsymbol{x}_t|t) - \frac{\boldsymbol{C}_t\boldsymbol{x}_0 - \boldsymbol{x}_t}{\sigma_t^2}\right|\right|^2\right]. \tag{7}$$

As shown, the objective of the model is to predict the (normalized) difference, $\boldsymbol{C}_t\boldsymbol{x}_0 - \boldsymbol{x}_t$, which is actually the noise, $\boldsymbol{\eta}_t$. We argue that even though this objective is theoretically grounded, in many cases, it would not work in practice because we would need infinite samples to actually learn the vector-field $\nabla_{\boldsymbol{x}_t}\log q_t(\boldsymbol{x}_t)$ in a way that would allow sampling.

Assume that the corruption process is blurring (at different levels) paired with additive noise of small magnitude. The objective written in Eq. (7) learns the distributions of blurry (and slightly noisy images) by just removing noise. Hence, in practice we might only learn these distributions locally (around the blurry images) and hence we might not be able to reduce the blurriness. This point might be better understood after we present our Sampling Method in Section 3.2.

To account for this problem, we propose a network reparametrization which leverages that we know the linear corruption mechanism, $C_t$. Specifically, we propose the following parametrization:

$$\boldsymbol{s}_\theta(\boldsymbol{x}_t|t) = \frac{\boldsymbol{C}_t\boldsymbol{h}_\theta(\boldsymbol{x}_t|t) - \boldsymbol{x}_t}{\sigma_t^2}. \tag{8}$$

Crucially, the network incorporates the corruption process. The loss becomes:

$$L(t) = \frac{1}{2}\mathbb{E}_{(\boldsymbol{x}_0,\boldsymbol{x}_t)\sim q_0(\boldsymbol{x}_0)q_t(\boldsymbol{x}_t|\boldsymbol{x}_0)}\frac{1}{\sigma_t^4}\left[||\boldsymbol{C}_t(\boldsymbol{h}_\theta(\boldsymbol{x}_t|t) - \boldsymbol{x}_0)||^2\right]. \tag{9}$$

When $C_t$ is a blurring matrix, this loss function is the MSE between the blurred prediction of $h_\theta$ and the blurred clean image. Finally, as observed in previous works (Song & Ermon, 2019; Ho et al., 2020; Karras et al., 2022), the optimization landscape becomes smoother when we are predicting the residual, instead of the clean image directly. This corresponds to the additional reparametrization:

$$\boldsymbol{h}_\theta(\boldsymbol{x}_t|t) = \boldsymbol{\phi}_\theta(\boldsymbol{x}_t|t) + \boldsymbol{x}_t, \tag{10}$$

which leads to the final form of our loss function:

$$L(t) = \frac{1}{2}\mathbb{E}_{(\boldsymbol{x}_0,\boldsymbol{x}_t)\sim q_0(\boldsymbol{x}_0)q_t(\boldsymbol{x}_t|\boldsymbol{x}_0)}\frac{1}{\sigma_t^4}\left[||\boldsymbol{C}_t\left(\boldsymbol{\phi}_\theta(\boldsymbol{x}_t|t) - \boldsymbol{r}_t\right)||^2\right], \tag{11}$$

where $\boldsymbol{r}_t$ is the residual with respect to the clean image, i.e. $\boldsymbol{r}_t = \boldsymbol{x}_0 - \boldsymbol{x}_t$. Following prior work, we use a single network conditioned on time $t$ that is optimized for all $L(t)$. Hence, the total loss is:

$$L = \mathbb{E}_{t\sim\mathcal{U}[0,1]}w(t)\left[\mathbb{E}_{(\boldsymbol{x}_0,\boldsymbol{x}_t)\sim q_0(\boldsymbol{x}_0)q_t(\boldsymbol{x}_t|\boldsymbol{x}_0)}\left[||\boldsymbol{C}_t\left(\boldsymbol{\phi}_\theta(\boldsymbol{x}_t|t) - \boldsymbol{r}_t\right)||^2\right]\right], \tag{12}$$

where the weights are usually chosen to be 1 or $1/\sigma_t^2$ (Karras et al., 2022; Kingma et al., 2021).

We call our training objective **Soft Score Matching**. The name is inspired from "soft filtering" a term used in photography to denote an image filter that removes fine details (e.g., blur, fading, etc). As in the Denoising Score Matching, the network is essentially trained to predict the residual to the clean image, but in our case, the loss is in the filtered space. When there is no filtering matrix, i.e. $C_t = I$, we recover the DSM objective used in (Song & Ermon, 2019; 2020; Song et al., 2021b).

**Comparison with objectives used in other works.** Previous (Anonymous, 2022) or concurrent (Bansal et al., 2022; Rissanen et al., 2022; Hoogeboom & Salimans, 2022) works that consider different degradations than Gaussian Diffusion, use the heuristic objective of predicting the clean image, i.e. they minimize: $||\boldsymbol{\phi}_\theta(\boldsymbol{x}_t|t) - \boldsymbol{r}_t||$. This is actually an upper-bound on our loss, i.e. $||\boldsymbol{C}_t\left(\boldsymbol{\phi}_\theta(\boldsymbol{x}_t|t) - \boldsymbol{r}_t\right)|| \leq ||\boldsymbol{C}_t||||\boldsymbol{\phi}_\theta(\boldsymbol{x}_t|t) - \boldsymbol{r}_t||$. Since the spectral norm, $||\boldsymbol{C}_t||$, is fixed, one can optimize for the *upper-bound* by minimizing $||\boldsymbol{\phi}_\theta(x_t|t) - \boldsymbol{r}_t||$. Instead, Soft Score Matching optimizes directly for learning the score. Experimentally, Soft Score Matching outperforms (ours FID: 1.85, theirs: 5.91), under the exact same setting, this simple baseline (see Experiments).

## 3.2 SAMPLING

**Naive Sampler.** Once the model is trained, we need a way to generate samples. The simplest idea is to use recursively our model, $\phi_\theta(x_t|t)$, to get estimates of the clean image, $\hat{x}_0$. To move from corruption level $t$ to corruption level $t - \Delta t$, we feed the image $\boldsymbol{x}_t$ to the model to get an estimate of the clean image, $\hat{x}_0$, and then corrupt back to level $t - \Delta t$. This idea is shown in Algorithm 1.

**Algorithm 1** Naive Sampler

---
**Require:** $p_1, \phi_\theta, \boldsymbol{C}_t, \sigma_t, \Delta t$
  $x_1 \sim p_1(\boldsymbol{x})$
  **for** $t = 1$ **to** $0$ **with step** $-\Delta t$ **do**
    $\hat{x}_0 = \phi_\theta(\boldsymbol{x}_t|t) + \boldsymbol{x}_t$
    $\boldsymbol{\eta}_t \sim \mathcal{N}(\boldsymbol{0}, \boldsymbol{I})$
    $\boldsymbol{x}_{t-\Delta t} \leftarrow \boldsymbol{C}_{t-\Delta t}\hat{x}_0 + \sigma_{t-\Delta t}\boldsymbol{\eta}_t$
  **end for**
  **return** $x_0$

---

**Momentum Sampler.** As we will show in the Experiments section, the naive sampler we presented above leads to generated images that lack diversity. We propose an simple, yet novel, alternative method for sampling from the general linear diffusion model presented in Eq. (4). Our method is inspired by the continuous formulation of diffusion models that is introduced in Song et al. (2021b).

The first step is to find a Markovian stochastic process that is "close" to the non-Markovian corruption process of Eq. (4). Consider the following SDE:

$$d\boldsymbol{x}_t = \dot{\boldsymbol{C}}_t \mathbb{E}[\boldsymbol{x}_0|\boldsymbol{x}_t]dt + \sqrt{\frac{d(\sigma_t^2)}{dt}}d\boldsymbol{w}, \tag{13}$$

where $\boldsymbol{w}$ is the standard Wiener process. This is a special case of the Itô SDE: $d\boldsymbol{x} = \boldsymbol{f}(\boldsymbol{x}, t)dt + g(t)d\boldsymbol{w}$, that appears in Song et al. (2021b), i.e. $\boldsymbol{f}(\boldsymbol{x}, t) = \dot{\boldsymbol{C}}_t\mathbb{E}[\boldsymbol{x}_0|\boldsymbol{x}_t]$ and $g(t) = \sqrt{\frac{d(\sigma_t^2)}{dt}}$. Crucially, $\mathbb{E}[\boldsymbol{x}_0|\boldsymbol{x}_t]$ does not depend on previous values of $\boldsymbol{x}_t$ and our SDE is indeed an Ito SDE.

To build some intuition, it is useful to think of the toy setting where the dataset contains one image, $\boldsymbol{\alpha} \in \mathbb{R}^n$. Under this setting, the Markovian corruption described by the SDE of Eq. (13) has the same marginal distributions as the process of Eq. (4). As we explain in the Appendix (Section E), in the general case, the SDE (13) introduces an approximation error with respect to the corruption process of Eq. (4) – the former uses the conditional expectation, $\mathbb{E}[\boldsymbol{x}_0|\boldsymbol{x}_t]$, while the latter corrupts directly $\boldsymbol{x}_0$. Nevertheless, the approximation error for the considered corruptions seems to be small, given the experimental success of the derived sampler. Intuitively, this is because for low corruption levels, the conditional expectation and $\boldsymbol{x}_0$ are close, while for high corruption levels, the distance is contracted by the multiplication with the corruption matrix $\boldsymbol{C}_t$ (see also Appendix, Section E).

Eq. (13) describes a reversible diffusion process (Anderson, 1982). The reverse is also an Ito SDE:

$$d\boldsymbol{x}_t = \left[\dot{\boldsymbol{C}}_t\mathbb{E}[\boldsymbol{x}_0|\boldsymbol{x}_t] - \frac{d(\sigma_t^2)}{dt}\nabla_{\boldsymbol{x}_t}\log q_t(\boldsymbol{x}_t)\right]dt + \sqrt{\frac{d(\sigma_t^2)}{dt}}d\bar{\boldsymbol{w}}. \tag{14}$$

where $\bar{\boldsymbol{w}}$ is a standard Wiener process when time flows backwards from $t = 1$ to $t = 0$. In practice, to solve Eq. (14), we discretize the SDE (i.e., apply Euler-Maruyama, and approximate the function derivatives with finite differences). The Euler-Maruyama discretization is given below:

$$\boldsymbol{x}_{t-\Delta t} - \boldsymbol{x}_t = (\boldsymbol{C}_{t-\Delta t} - \boldsymbol{C}_t)\mathbb{E}[\boldsymbol{x}_0|\boldsymbol{x}_t] - (\sigma_{t-\Delta t}^2 - \sigma_t^2)\nabla_{\boldsymbol{x}_t}\log q_t(\boldsymbol{x}_t) + \sqrt{\sigma_t^2 - \sigma_{t-\Delta t}^2}\boldsymbol{\eta}, \tag{15}$$

where $\boldsymbol{\eta} \sim N(\boldsymbol{0}, \boldsymbol{I})$. In this update Equation, there are two unknowns: the conditional expectation, $\mathbb{E}[\boldsymbol{x}_0|\boldsymbol{x}_t]$, and the score function, $\nabla_{\boldsymbol{x}_t}\log q_t(\boldsymbol{x}_t)$. We show that these are actually connected, through the Tweedie's (Efron, 2011; Robbins, 1956; Stein, 1981) formula. Specifically, it holds that:

$$\boldsymbol{C}_t\mathbb{E}[\boldsymbol{x}_0|\boldsymbol{x}_t] = \boldsymbol{x}_t + \sigma_t^2\nabla_{\boldsymbol{x}_t}\log q_t(\boldsymbol{x}_t). \tag{16}$$

The proof is given for completeness in the Appendix, Lemma A.1. To estimate $\nabla_{\boldsymbol{x}_t}\log q_t(\boldsymbol{x}_t)$ we use our model that provably learns the score according to Theorem 3.1. Putting everything together:

$$\Delta\boldsymbol{x}_t = \boldsymbol{x}_{t-\Delta t} - \boldsymbol{x}_t = \tag{17}$$

$$= (\boldsymbol{C}_{t-\Delta t} - \boldsymbol{C}_t)\underbrace{\left(\phi_\theta(\boldsymbol{x}_t|t) + \boldsymbol{x}_t\right)}_{\hat{x}_0} - \left(\frac{\sigma_{t-\Delta t}^2 - \sigma_t^2}{\sigma_t^2}\right)\left(\boldsymbol{C}_t\left(\phi_\theta(\boldsymbol{x}_t|t) + \boldsymbol{x}_t\right) - \boldsymbol{x}_t\right) + \sqrt{\sigma_t^2 - \sigma_{t-\Delta t}^2}\boldsymbol{\eta}.$$

Our sampler is summarized in Algorithm 2. Essentially, there is one update for deblurring and one for denoising. At the core of this update equation, is the prediction of the clean image, $\hat{x}_0$. Once

---

**Algorithm 2** Momentum Sampler

---

**Require:** $p_1, \phi_\theta, \boldsymbol{C}_t, \sigma_t, \Delta t$
  $\boldsymbol{x}_1 \sim p_1(\boldsymbol{x})$
  **for** $t = 1$ **to** $0$ **with step** $-\Delta t$ **do**
    $\hat{\boldsymbol{x}}_0 = \phi_\theta(\boldsymbol{x}_t|t) + \boldsymbol{x}_t$                            $\triangleright$ Coarse prediction of the clean image.
    $\hat{\boldsymbol{y}}_t \leftarrow \boldsymbol{C}_t \hat{\boldsymbol{x}}_0$                               $\triangleright$ Coarse prediction of filtered image at $t$.
    $\boldsymbol{\eta}_t \sim \mathcal{N}(\boldsymbol{0}, \boldsymbol{I})$
    $\hat{\boldsymbol{\epsilon}}_t \leftarrow \hat{\boldsymbol{y}}_t - \boldsymbol{x}_t$                                   $\triangleright$ Estimate of noise at $t$.
    $\boldsymbol{z}_{t-\Delta t} \leftarrow \boldsymbol{x}_t - \frac{(\sigma_{t-\Delta t}^2 - \sigma_t^2)}{\sigma_t^2} \hat{\boldsymbol{\epsilon}}_t + \sqrt{\sigma_t^2 - \sigma_{t-\Delta t}^2} \boldsymbol{\eta}_t$     $\triangleright$ Filtered image at $t$ with noise at $t - \Delta t$.
    $\hat{\boldsymbol{y}}_{t-\Delta t} \leftarrow \boldsymbol{C}_{t-\Delta t} \hat{\boldsymbol{x}}_0$                    $\triangleright$ Coarse prediction of filtered image at $t - \Delta t$.
    $\boldsymbol{x}_{t-\Delta t} \leftarrow \boldsymbol{z}_{t-\Delta t} + (\hat{\boldsymbol{y}}_{t-\Delta t} - \hat{\boldsymbol{y}}_t)$           $\triangleright$ Filtered image at $t - \Delta t$ with noise at $t - \Delta t$.
  **end for**
  **return** $\boldsymbol{x}_0$

---

the clean image is predicted, we blur it back to two different corruption levels, $t$ and $t - \Delta t$. The deblurring gradient is the residual between the blurred images at levels $t - \Delta t$ and $t$. Interestingly, the denoising update is the same as the one used in typical score-based models (that only use additive noise). In fact, if there is no blur ($\boldsymbol{C}_t = I$), our sampler becomes exactly the sampler used for the Variance Exploding (VE) SDE in Song et al. (2021b).

We call our sampler **Momentum Sampler** because we can think of it as a generalization of the update of the Naive Sampler, where there is a momentum term. To understand this better, we look at the setting where there is no noise. Then, the update rule of the Momentum Sampler is:

$$\Delta \boldsymbol{x}_t = \boldsymbol{C}_{t-\Delta t} \hat{\boldsymbol{x}}_0 - \boldsymbol{C}_t \hat{\boldsymbol{x}}_0. \tag{18}$$

As seen, the first term is what the Naive Sampler would use to update the image at level $t$ and the second term is what the Naive Sampler would use to update the image at level $t - \Delta t$. If these two directions are aligned, then the gradient $\Delta \boldsymbol{x}_t$ is small. Hence, there is a notion of momentum, analogous to how the term is used in classical optimization. In the Appendix, Section B.2, we also present a DDIM-type (Song et al., 2021a) sampler for which the momentum term also appears.

**Probability Flow Momentum Sampler.** The update rule of our sampler was derived using the discretization of the backward SDE of our corruption, given in Eq. (14). Similarly to Song et al. (2021b), we can also consider the Ordinary Differential Equation (ODE) associated with this SDE:

$$d\boldsymbol{x}_t = \left[ \dot{\boldsymbol{C}}_t \mathbb{E}[\boldsymbol{x}_0 | \boldsymbol{x}_t] - \frac{1}{2} \frac{d(\sigma_t^2)}{dt} \nabla_{\boldsymbol{x}_t} \log q_t(\boldsymbol{x}_t) \right] dt. \tag{19}$$

Again, we can approximate $\dot{\boldsymbol{C}}_t \mathbb{E}[\boldsymbol{x}_0 | \boldsymbol{x}_t], \nabla_{\boldsymbol{x}_t} \log q_t(\boldsymbol{x}_t)$ with our trained network and get a deterministic version of the Momentum Sampler, which we call Probability Flow Momentum Sampler, as in Song et al. (2021b). We detail our derivations in the Appendix, Section B.1.

### 3.3 SCHEDULING

The last piece of our framework is how to choose the corruption levels, i.e. the scheduling of the diffusion. For example, for Gaussian Blur, the scheduling decides how much time is spent in the diffusion in the very blurry, somewhat blurry and almost no blurry regimes. We provide a principled way of choosing the corruption levels for arbitrary corruption processes.

Let $\mathcal{D}_0$ the distribution of real images and $\mathcal{D}_1$ be a known distribution that we know how to sample from, e.g. a Normal Distribution. In the design phase of score-based modeling, the goal is to choose a set of intermediate distributions, $\{\mathcal{D}_t\}$, that smoothly transform images from $\mathcal{D}_0$ to samples from the distribution $\mathcal{D}_1$. Let $\Theta = \{\theta_1, \theta_2, ..., \theta_k, ...\}$ be the space of diffusion parameters, i.e. each $\theta_i$ corresponds to a distribution $\mathcal{D}_{\theta_i}$. In the case of blur for example, $\theta_i$ controls how much we blur the image. Let also $\mathcal{M} : \mathcal{X} \times \mathcal{X} \rightarrow \mathbb{R}$ be a metric that measures distances between distributions, e.g. $\mathcal{M}$ might be the Wasserstein Distance of distributions with support $\mathcal{X}$.

We construct a weighted graph $G_\epsilon$ with the nodes being the distributions and the weights given by:

$$w\left(\mathcal{D}_{\theta_i}, \mathcal{D}_{\theta_j}\right) = \begin{cases} \mathcal{M}\left(\mathcal{D}_{\theta_i}, \mathcal{D}_{\theta_j}\right), & \text{if } \mathcal{M}\left(\mathcal{D}_{\theta_i}, \mathcal{D}_{\theta_j}\right) \leq \epsilon, \\ \infty, & \text{otherwise.} \end{cases} \tag{20}$$

For a fixed $\epsilon$, we *choose the distributions that minimize the cost of the path between $\mathcal{D}_0$ and $\mathcal{D}_1$.* The parameter $\epsilon$ expresses the power of the best neural network we can train to reverse one step of the diffusion. If $\epsilon = \infty$, then for any metric $M$, the shortest path is to go directly from $D_0$ to $D_1$. However, it is impossible to denoise in one step a super noisy image. Hence, we need to go through many intermediate steps, which is forced by setting $\epsilon$ to a smaller value. As we increase the number of the candidate distributions we are getting closer to finding the geodesic between $D_0$ and $D_1$. However, the computational cost of the method increases since we need to estimate all the pairwise distances $\mathcal{M}(D_{\theta_i}, D_{\theta_j})$. In practice, we use a relatively small number of candidate distributions, e.g. $T = 256$ and once the path is found, we do linear interpolation to extend to the continuous case.

## 4 EXPERIMENTS

We evaluate our method in CelebA-64 and CIFAR-10. We show that by just changing the corruption mechanism (and using our framework for scheduling, learning and sampling) we significantly improve the FID score and reduce the sampling time. We use the architecture and the training hyperparameters from Song et al. (2021b) (full details can be found in the Appendix).

For most of our experiments, we use Gaussian Blur as our primary corruption mechanism. To illustrate that our method works more generally, we also show results with masking which is discussed separately later. Consistent with the description of our method, our deterministic corruptions are also paired with additive low magnitude noise. This is required by our theoretical results, otherwise the conditional log-likelihood, $\log q(\boldsymbol{x}_t | \boldsymbol{x}_0)$ would be undefined. We also find the addition of noise beneficial in practice (see Appendix F.1.1 for ablation studies on the role of noise).

**Scheduling.** We use the standard geometric scheduling for the noise (Song & Ermon, 2019; 2020; Song et al., 2021b) and use the methodology described in Section 3.3 to select the blur levels. We use the Wasserstein distance as the metric $\mathcal{M}$ to measure how close are the different distributions. To clearly illustrate the switch to a different corruption, our diffusion has an initial stage that increases the noise (with no blur) and then we fix the noise (to a small value, e.g. $\sigma = 0.1$) and change (using our scheduling framework) the amount of blur. Our diffusion spends less than $20\%$ of the total time in the initial stage that increases the noise. We ablate those choices in Section F.1 of the Appendix.

One important property of our framework is that the scheduling is dataset specific. Intuitively, the way we corrupt depends on the nature of the data we are modelling. The interested reader can find the found schedulings for each dataset in Figure 6 of the Appendix.

**Results.** We train our networks on CelebA-64 and CIFAR-10 using the found schedulings and the training objective of Eq. (12). We start by showing uncurated samples of our models in Figure 2.

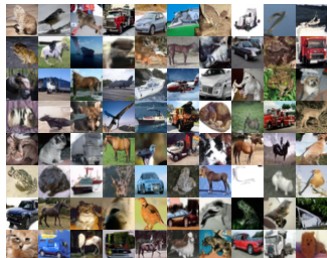 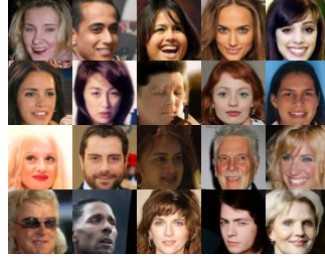

Figure 2: Uncurated samples from our trained models on CIFAR-10 (left) and CelebA (right).

The generated images have high diversity and fidelity in both datasets. We compare the FID (Heusel et al., 2017) obtained by our method on CelebA with many natural baselines that use any of the VE, VP or subVP SDEs. Specifically, we compare against DDPM (Ho et al., 2020) that uses the VP SDE, DDIM (Song et al., 2021a) that uses the same model but with a different sampler, DDPM++ (Kim et al., 2022b) that is the state-of-the-art model for VP-SDE and the NCSN++ models (Song et al., 2021b) trained with the VE and subVP-SDEs. For a fair comparison, we only use reported numbers in published papers for the baselines and we do not rerun them ourselves. *Our model achieves state-of-the-art FID score, 1.85, in CelebA,* outperforming all the other methods. The results are summarized in Table 1. For CIFAR-10, we obtain FID score 3.86 with our Probability Flow Momentum Sampler and 3.91 with our Momentum Sampler. We summarize the results in Table 2. Our

best FID is competitive with similar samplers to similar methods, e.g. with NCSN++ (VE SDE) with Reverse SDE sampling. However, it is behind other state-of-the-art models such as LSGM (Vahdat et al., 2021) and DDPM (Ho et al., 2020). We believe that this performance gap can be decreased in the future by further research in the area of diffusion models with general corruptions.

Our method is superior in *sampling time*, for both CIFAR-10 and CelebA. Figure 3 shows how FID changes based on the Number of Function Evaluations (NFEs). Our method requires significant less steps to achieve the same or better quality than NCSN++ (VE SDE) (Song et al., 2021b), using the same architecture and training hyperparameters (FID values taken from (Ma et al., 2022)).

| Model | FID |
|---|---|
| DDPM (VP SDE) (Ho et al., 2020) | 3.26 |
| DDIM (VP SDE) (Song et al., 2021a) | 3.51 |
| DDPM++ (VP SDE) (Kim et al., 2022b) | 1.90 |
| NCSN++ (subVP-SDE) (Song et al., 2021b) | 3.95 |
| NCSN++ (VE SDE) (Song et al., 2021b) | 3.25 |
| **Ours** (VE SDE + Blur) | **1.85** |

Table 1: FID results on CelebA-64.

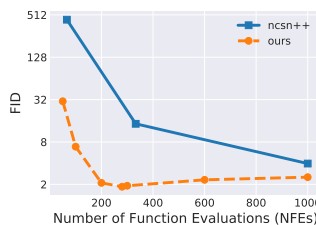

Figure 3: FID versus NFEs (CelebA-64).

**Ablation Study for Sampling.** For all the results we presented so far, we used the Momentum Sampler that we introduced in Section 3.2 and Algorithm 2. In this section, we ablate the choice of the sampler. Specifically, we compare with the intuitive Naive Sampler described in Algorithm 1. We show that the choice of the sampler has a dramatic effect in the quality and the diversity of the generated images. Results are shown in Figure 4. The images from the Naive Sampler seem repetitive and lack details. This is reflected in the poor FID score. The Momentum Sampler leads to images with greater variety and detail, dramatically improving FID from 27.82 to 1.85.

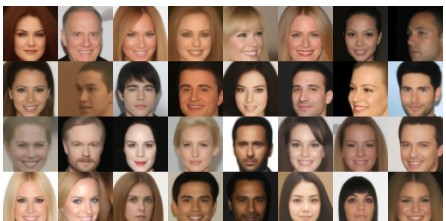

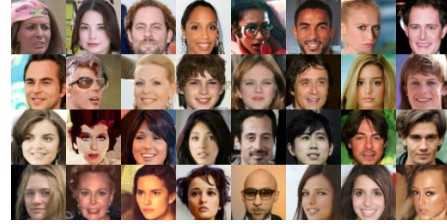

(a) Naive Sampler (uncurated). FID: 27.82.  (b) Momentum Sampler (uncurated). FID: **1.85**.

Figure 4: Effect of sampling method on the quality of the samples. Naive Sampler (4a) gives repetitive images that lack details. Momentum Sampler (4b) dramatically improves the sampling quality and the FID score.

**Training Objective.** Concurrent works (Hoogeboom & Salimans, 2022; Bansal et al., 2022) that also consider different corruption processes have used as their loss a simple objective: given an input image, they predict the residual to the clean. As explained in Section 3.1, this is equivalent to optimizing for an upper-bound of the Soft Score Matching objective (which is minimizing the error to the score-function, see Theorem 3.1). We show that if we use our exact pipeline (model, scheduling, sampler, training hyperparameters and so on) and we replace our Soft Score matching Loss with the loss introduced in Bansal et al. (2022), FID score increases from 1.85 to 5.91. This experiment shows the effectiveness of the Soft Score Matching objective.

**Ablation Studies for Scheduling.** We perform extensive ablations on the scheduling to understand the role of noise in the framework and whether our scheme outperforms other natural baselines. The results are detailed in the Appendix, Section F.1. Our main findings are: i) the Momentum Sampler works even if noise and blur are changing simultaneously (i.e. for schedulings with non-fixed noise), ii) lowering the maximum value of noise leads to important performance degradation – for very small noise the method completely fails and iii) our found scheduling outperforms significantly (baseline FID: 8.35, ours: 1.85) a natural baseline that sets blur parameters such that MSE between the corrupted and the clean image decays in the same rate for Gaussian Denoising and Soft Diffusion.

**Masking Diffusion Models.** To show the generality of our framework, we also train models with (discrete) masking diffusion paired with noise. Figure 1 (top 2 rows) shows the forward and the (learned) reverse process for blur on the left and masking on the right. We train the model with our Soft Score Matching objective on CelebA. Unconditional samples from the model trained with masking can be found in Figure 13 of the Appendix. In Figure 5, we show the predictions of our two trained models (blur and masking) for the conditional mean, $\mathbb{E}[\boldsymbol{x}_0|\boldsymbol{x}_t]$, at different times of the diffusion. Soft Score Matching trains the model to make a prediction that matches the real images in the filtered space. Hence, given masked images the masking model is incentivized to predict right only the observed noisy region. As diffusion time $t$ becomes smaller (cleaner images), the observed region grows and the model predicts bigger windows. Although it is interesting that we can train Masking Diffusion models, there are several limitations compared to Blur Diffusion (and even Gaussian Diffusion). We observe that these models are very slow to sample from: with $1024$ sampling steps, FID is $30.92$ while with $4096$, FID improves to $12.37$.

## 5 RELATED WORK

We showed that by just changing the corruption mechanism, we observe important computational benefits. Reducing the number of function evaluations for sample generation with diffusion models is an active area of research. Jolicoeur-Martineau et al. (2021); Liu et al. (2022); Song et al. (2021a) propose more efficient samplers for diffusion models. Rombach et al. (2022); Daras et al. (2022) train diffusion models on low-dimensional latent spaces. Xiao et al. (2022) combine diffusion models with Generative Adversarial Networks (GANs) to allow for bigger denoising steps. Ryu & Ye (2022); Ho et al. (2022a) generate high-resolution images progressively, from coarse to fine quality. Salimans & Ho (2022) train progressively a student network that mimics the teacher diffusion model with fewer sampling steps. We note that all these works are orthogonal to ours and therefore can be used in combination with our framework for even faster sampling.

On scheduling, there is closely related work by Bao et al. (2022). The authors find a closed form solution (w.r.t. the score function) for the variance of the reverse SDE for Gaussian Diffusion. Then, they select a noise scheduling that minimizes the KL along the path from the initial to the final distribution. In our work, we use Wasserstein distances and consider general corruption processes for which it is unclear whether such a closed form solution exists. Instead, we estimate the distances in a data-driven way. This allows us to schedule arbitrary diffusion processes in a principled way.

There is significant recent (Anonymous, 2022) and concurrent work (Rissanen et al., 2022; Bansal et al., 2022; Hoogeboom & Salimans, 2022) that proposes diffusion with other degradations. These works have significant differences since they use different loss functions and sampling methods. Soft Score Matching experimentally outperforms (under the exact same setting) the loss functions used in the concurrent works (ours FID: $1.85$, theirs: $5.91$). Our (blur) models obtain state-of-the-art FID for linear diffusion models on CelebA (FID $1.85$). For CIFAR10, we outperform (FID: $3.86$) Gaussian diffusion with Variance Exploding SDE (Song et al., 2021b). Hoogeboom & Salimans (2022) also use blurring (but with Variance Preserving SDE) to further push the CIFAR-10 performance to FID $3.17$. These advancements show that there are multiple diffusions with promising potential.

## 6 CONCLUSIONS AND FUTURE WORK

We presented a framework to train and sample from diffusion models that reverse general linear corruptions. We showed that by changing the corruption process, we can get significant sample quality improvements and computational benefits. This work opens several future research directions. For example, it is possible to optimize or learn the corruption process for solving a specific type of inverse problem. It is also worth exploring if mixing different corruptions (blur, noise, masking, etc.) improves performance. Our work could also be extended to the non-linear case, leveraging the techniques introduced in (Rombach et al., 2022; Kim et al., 2022a; Wang et al., 2022). Finally, it is important to understand the role of noise, from both a theoretical and practical standpoint.

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

## A APPENDIX

### A.1 PROOFS

**Theorem 3.1.** *Let $q_0, q_t$ be two distributions in $\mathbb{R}^n$. Assume that all the conditional distributions, $q_t(\boldsymbol{x}_t|\boldsymbol{x}_0)$, are supported and differentiable in $\mathbb{R}^n$. Let:*

$$J_1(\theta) = \frac{1}{2}\mathbb{E}_{\boldsymbol{x}_t \sim q_t}\left[||\boldsymbol{s}_\theta(\boldsymbol{x}_t) - \nabla_{\boldsymbol{x}_t}\log q_t(\boldsymbol{x}_t)||^2\right], \tag{21}$$

$$J_2(\theta) = \frac{1}{2}\mathbb{E}_{(\boldsymbol{x}_0,\boldsymbol{x}_t)\sim q_0(\boldsymbol{x}_0)q_t(\boldsymbol{x}_t|\boldsymbol{x}_0)}\left[||\boldsymbol{s}_\theta(\boldsymbol{x}_t) - \nabla_{\boldsymbol{x}_t}\log q_t(\boldsymbol{x}_t|\boldsymbol{x}_0)||^2\right]. \tag{22}$$

*Then, there is a universal constant $C$ (that does not depend on $\theta$) such that: $J_1(\theta) = J_2(\theta) + C$.*

The proof of this Theorem is following the calculations of Vincent (2011). We observe that as long as the technical conditions listed are satisfied, the proof holds independent of the corruption type. We provide the proof below for the shake of completeness.

*Proof of Theorem 3.1.*
$$J_1(\theta) = \frac{1}{2}\mathbb{E}_{\boldsymbol{x}_t \sim q_t}\left[||\boldsymbol{s}_\theta(\boldsymbol{x}_t)||^2 - 2\boldsymbol{s}_\theta(\boldsymbol{x}_t)^T\nabla_{\boldsymbol{x}_t}\log q_t(\boldsymbol{x}_t) + ||\nabla\boldsymbol{x}_t\log q_t(\boldsymbol{x}_t)||^2\right] \tag{23}$$

$$= \frac{1}{2}\mathbb{E}_{\boldsymbol{x}_t \sim q_t}\left[||\boldsymbol{s}_\theta(\boldsymbol{x}_t)||^2\right] - \mathbb{E}_{\boldsymbol{x}_t \sim q_t}\left[s_\theta(\boldsymbol{x}_t)^T\nabla_{\boldsymbol{x}_t}\log q_t(\boldsymbol{x}_t)\right] + C_1. \tag{24}$$

Similarly,

$$J_2(\theta) = \frac{1}{2}\mathbb{E}_{\boldsymbol{x}_t \sim q_t}\left[||\boldsymbol{s}_\theta(\boldsymbol{x}_t)||^2\right] - \mathbb{E}_{(\boldsymbol{x}_0,\boldsymbol{x}_t)\sim q_0(\vec{x}_0)q_t(\boldsymbol{x}_t|\boldsymbol{x}_0)}\left[\boldsymbol{s}_\theta(\boldsymbol{x}_t)^T\nabla_{\boldsymbol{x}_t}\log q_t(\boldsymbol{x}_t|\boldsymbol{x}_0)\right] + C_2. \tag{25}$$

It suffices to show that:

$$\mathbb{E}_{\boldsymbol{x}_t \sim q_t}\left[s_\theta(\boldsymbol{x}_t)^T\nabla_{\boldsymbol{x}_t}\log q_t(\boldsymbol{x}_t)\right] = \mathbb{E}_{(\boldsymbol{x}_0,\boldsymbol{x}_t)\sim q_0(\boldsymbol{x}_0)q_t(\boldsymbol{x}_t|\boldsymbol{x}_0)}\left[s_\theta(\boldsymbol{x}_t)^T\nabla_{\boldsymbol{x}_t}\log q_t(\boldsymbol{x}_t|\boldsymbol{x}_0)\right]. \tag{26}$$

We start with the second term.

$$\mathbb{E}_{(\boldsymbol{x}_0,\boldsymbol{x}_t)\sim q_0(\boldsymbol{x}_0)q_t(\boldsymbol{x}_t|\boldsymbol{x}_0)}\left[\boldsymbol{s}_\theta(\boldsymbol{x}_t)^T\nabla_{\boldsymbol{x}_t}\log q_t(\boldsymbol{x}_t|\boldsymbol{x}_0)\right]$$

$$= \int_{\boldsymbol{x}_0}\int_{\boldsymbol{x}_t} q_0(\boldsymbol{x}_0)q_t(\boldsymbol{x}_t|\boldsymbol{x}_0)\boldsymbol{s}_\theta(\boldsymbol{x}_t)^T\nabla_{\boldsymbol{x}_t}\log q_t(\boldsymbol{x}_t|\boldsymbol{x}_0)\mathrm{d}\boldsymbol{x}_t\mathrm{d}\boldsymbol{x}_0 \tag{27}$$

$$= \int_{\boldsymbol{x}_0}\int_{\boldsymbol{x}_t} \boldsymbol{s}_\theta^T(\boldsymbol{x}_t)\left(q_0(\boldsymbol{x}_0)q_t(\boldsymbol{x}_t|\boldsymbol{x}_0)\nabla_{\boldsymbol{x}_t}\log q_t(\boldsymbol{x}_t|\boldsymbol{x}_0)\right)\mathrm{d}\boldsymbol{x}_t\mathrm{d}\boldsymbol{x}_0 \tag{28}$$

$$= \int_{\boldsymbol{x}_0}\int_{\boldsymbol{x}_t} \boldsymbol{s}_\theta^T(\boldsymbol{x}_t)\left(q_0(\boldsymbol{x}_0)q_t(\boldsymbol{x}_t|\boldsymbol{x}_0)\frac{1}{q_t(\boldsymbol{x}_t|\boldsymbol{x}_0)}\nabla_{\boldsymbol{x}_t}q_t(\boldsymbol{x}_t|\boldsymbol{x}_0)\right)\mathrm{d}\boldsymbol{x}_t\mathrm{d}\boldsymbol{x}_0 \tag{29}$$

$$= \int_{\boldsymbol{x}_0}\int_{\boldsymbol{x}_t} \boldsymbol{s}_\theta^T(\boldsymbol{x}_t)\left(q_0(\boldsymbol{x}_0)\nabla_{\boldsymbol{x}_t}q_t(\boldsymbol{x}_t|\boldsymbol{x}_0)\right)\mathrm{d}\boldsymbol{x}_t\mathrm{d}\boldsymbol{x}_0 \tag{30}$$

$$= \int_{\boldsymbol{x}_t}\int_{\boldsymbol{x}_0} \boldsymbol{s}_\theta^T(\boldsymbol{x}_t)\left(q_0(\boldsymbol{x}_0)\nabla_{\boldsymbol{x}_t}q_t(\boldsymbol{x}_t|\boldsymbol{x}_0)\right)\mathrm{d}\boldsymbol{x}_0\mathrm{d}\boldsymbol{x}_t \tag{31}$$

$$= \int_{\boldsymbol{x}_t} \boldsymbol{s}_\theta^T(\boldsymbol{x}_t)\left(\int_{\boldsymbol{x}_0} q_0(\boldsymbol{x}_0)\nabla_{\boldsymbol{x}_t}q_t(\boldsymbol{x}_t|\boldsymbol{x}_0)\mathrm{d}\boldsymbol{x}_0\right)\mathrm{d}\boldsymbol{x}_t \tag{32}$$

$$= \int_{\boldsymbol{x}_t} \boldsymbol{s}_\theta^T(\boldsymbol{x}_t)\left(\int_{\boldsymbol{x}_0} \nabla_{\boldsymbol{x}_t}\left(q_0(\boldsymbol{x}_0)q_t(\boldsymbol{x}_t|\boldsymbol{x}_0)\right)\mathrm{d}\boldsymbol{x}_0\right)\mathrm{d}\boldsymbol{x}_t \tag{33}$$

$$= \int_{\boldsymbol{x}_t} \boldsymbol{s}_\theta^T(x_t)\left(\nabla_{\boldsymbol{x}_t}\left(\int_{\boldsymbol{x}_0} q_0(\boldsymbol{x}_0)q_t(\boldsymbol{x}_t|\boldsymbol{x}_0)\mathrm{d}\boldsymbol{x}_0\right)\right)\mathrm{d}\boldsymbol{x}_t \tag{34}$$

$$= \int_{\boldsymbol{x}_t} \boldsymbol{s}_\theta^T(\boldsymbol{x}_t)\nabla_{\boldsymbol{x}_t}q_t(\boldsymbol{x}_t)\mathrm{d}\boldsymbol{x}_t \tag{35}$$

$$= \int_{\boldsymbol{x}_t} q_t(\boldsymbol{x}_t)\boldsymbol{s}_\theta^T(\boldsymbol{x}_t)\nabla_{\boldsymbol{x}_t}\log q_t(\boldsymbol{x}_t)\mathrm{d}\boldsymbol{x}_t \tag{36}$$

$$= \mathbb{E}_{\boldsymbol{x}_t \sim q_t(\boldsymbol{x}_t)}\left[\boldsymbol{s}_\theta^T(\boldsymbol{x}_t)\nabla_{\boldsymbol{x}_t}\log q_t(\boldsymbol{x}_t)\right]. \tag{37}$$

$\square$

**Lemma A.1** (Tweedie's formula). *Consider the corruption process:* $\boldsymbol{x}_t = \boldsymbol{C}_t\boldsymbol{x}_0 + \sigma_t\boldsymbol{\eta}$*, where* $\boldsymbol{\eta} \sim \mathcal{N}(\boldsymbol{0},\boldsymbol{I})$*. Denote with* $q_t(\boldsymbol{x}_t)$ *the density of* $\boldsymbol{x}_t$ *and assume that* $\log q_t(\boldsymbol{x}_t)$ *is differentiable everywhere. Then, for the Minimum Mean-Square Estimation (MMSE) of* $\boldsymbol{x}_0$ *given* $\boldsymbol{x}_t$*, it holds that:*

$$\boldsymbol{C}_t\mathbb{E}[\boldsymbol{x}_0|\boldsymbol{x}_t] = \boldsymbol{x}_t + \sigma_t^2\nabla_{\boldsymbol{x}_t}\log q_t(\boldsymbol{x}_t). \tag{38}$$

*Proof.*

$$q_t(\boldsymbol{x}_t) = \int q_t(\boldsymbol{x}_t|\boldsymbol{x}_0)q_0(\boldsymbol{x}_0)\mathrm{d}\boldsymbol{x}_0 \Rightarrow \tag{39}$$

$$\nabla_{\boldsymbol{x}_t}q_t(\boldsymbol{x}_t) = \int q_0(\boldsymbol{x}_0)\nabla_{\boldsymbol{x}_t}q_t(\boldsymbol{x}_t|\boldsymbol{x}_0)\mathrm{d}\boldsymbol{x}_0 \tag{40}$$

$$= \int q_0(\boldsymbol{x}_0)q_t(\boldsymbol{x}_t|\boldsymbol{x}_0)\nabla_{\boldsymbol{x}_t}\log q_t(\boldsymbol{x}_t|\boldsymbol{x}_0)\mathrm{d}\boldsymbol{x}_0. \tag{41}$$

We know that: $q_t(\boldsymbol{x}_t|\boldsymbol{x}_0) = \mathcal{N}(x_t; \boldsymbol{\mu} = C_t\boldsymbol{x}_0, \boldsymbol{\Sigma} = \sigma_t^2\boldsymbol{I})$. Hence,

$$\nabla_{\boldsymbol{x}_t} q_t(\boldsymbol{x}_t) = \int q_0(\boldsymbol{x}_0)q_t(\boldsymbol{x}_t|\boldsymbol{x}_0)\frac{\boldsymbol{C}_t\boldsymbol{x}_0 - \boldsymbol{x}_t}{\sigma_t^2}\mathrm{d}\boldsymbol{x}_0 \tag{42}$$

$$= \frac{1}{\sigma_t^2}\left(\boldsymbol{C}_t\int q_0(\boldsymbol{x}_0)q_t(\boldsymbol{x}_t|\boldsymbol{x}_0)\boldsymbol{x}_0\mathrm{d}\boldsymbol{x}_0 - \boldsymbol{x}_t\int q_0(\boldsymbol{x}_0)q_t(\boldsymbol{x}_t|\boldsymbol{x}_0)\mathrm{d}\boldsymbol{x}_0\right) \tag{43}$$

$$= \frac{1}{\sigma_t^2}\left(\boldsymbol{C}_t\int q_0(\boldsymbol{x}_0|\boldsymbol{x}_t)q_t(\boldsymbol{x}_t)\boldsymbol{x}_0\mathrm{d}\boldsymbol{x}_0 - \boldsymbol{x}_tq_t(\boldsymbol{x}_t)\right) \Rightarrow \tag{44}$$

$$\frac{\nabla_{\boldsymbol{x}_t} q_t(\boldsymbol{x}_t)}{q_t(\boldsymbol{x}_t)} = \frac{1}{\sigma_t^2}\left(\boldsymbol{C}_t\mathbb{E}[\boldsymbol{x}_0|\boldsymbol{x}_t] - \boldsymbol{x}_t\right) \Leftrightarrow \tag{45}$$

$$\nabla_{\boldsymbol{x}_t} \log q_t(\boldsymbol{x}_t) = \frac{1}{\sigma_t^2}\left(\boldsymbol{C}_t\mathbb{E}[\boldsymbol{x}_0|\boldsymbol{x}_t] - \boldsymbol{x}_t\right). \tag{46}$$

$\square$

# B DETERMINISTIC SAMPLERS

## B.1 PROBABILITY FLOW ODE

In the main text, we derived our Momentum Sampler by analyzing the Backward SDE associated with our corruption process. Inspired by the works of Song et al. (2021b); Maoutsa et al. (2020), we also consider deterministic sampling that is derived by looking at the ODE that describes our diffusion. Particularly, the ODE:

$$\mathrm{d}\boldsymbol{x}_t = \left[\boldsymbol{f}(\boldsymbol{x}_t, t) - \frac{1}{2}g^2(t)\nabla_{\boldsymbol{x}_t}\log q_t(\boldsymbol{x}_t)\right]\mathrm{d}t, \tag{47}$$

has the same marginal distributions (Anderson, 1982; Maoutsa et al., 2020; Song et al., 2021b; Chen et al., 2018) with the Backward SDE:

$$\mathrm{d}\boldsymbol{x}_t = \left[\boldsymbol{f}(\boldsymbol{x}_t, t) - g^2(t)\nabla_{\boldsymbol{x}_t}\log q_t(\boldsymbol{x}_t)\right]\mathrm{d}t + g(t)\mathrm{d}\bar{\boldsymbol{w}}. \tag{48}$$

For our case, Eq. (47) becomes:

$$\mathrm{d}\boldsymbol{x}_t = \left[\dot{\boldsymbol{C}}_t\mathbb{E}[\boldsymbol{x}_0|\boldsymbol{x}_t] - \frac{1}{2}\frac{\mathrm{d}(\sigma_t^2)}{\mathrm{d}t}\nabla_{\boldsymbol{x}_t}\log q_t(\boldsymbol{x}_t)\right]\mathrm{d}t. \tag{49}$$

The first-order discretization of this ODE is given below:

$$\boldsymbol{x}_{t-\Delta t} - \boldsymbol{x}_t = (\boldsymbol{C}_{t-\Delta t} - \boldsymbol{C}_t)\mathbb{E}[\boldsymbol{x}_0|\boldsymbol{x}_t] - \frac{(\sigma_{t-\Delta t}^2 - \sigma_t^2)}{2}\nabla_{\boldsymbol{x}_t}\log q_t(\boldsymbol{x}_t). \tag{50}$$

We estimate $\boldsymbol{C}_t\mathbb{E}[\boldsymbol{x}_0|\boldsymbol{x}_t]$ and $\nabla_{\boldsymbol{x}_t}\log q_t(\boldsymbol{x}_t)$ with our neural network and we get the Neural ODE (Chen et al., 2018):

$$\Delta\boldsymbol{x}_t = \boldsymbol{x}_{t-\Delta t} - \boldsymbol{x}_t = \tag{51}$$

$$= (\boldsymbol{C}_{t-\Delta t} - \boldsymbol{C}_t)\left(\boldsymbol{\phi}_\theta(\boldsymbol{x}_t|t) + \boldsymbol{x}_t\right) - \frac{1}{2}\left(\frac{\sigma_{t-\Delta t}^2 - \sigma_t^2}{\sigma_t^2}\right)\left(\boldsymbol{C}_t\left(\boldsymbol{\phi}_\theta(\boldsymbol{x}_t|t) + \boldsymbol{x}_t\right) - \boldsymbol{x}_t\right),$$

which is the update rule of our Probability Flow ODE Momentum Sampler.

We note that our simple discretization is not the only way to solve the ODE of Eq. (47). We can use more sophisticated, e.g. see Dormand & Prince (1980).

## B.2 DDIM

A popular sampling scheme is the DDIM method, introduced in Song et al. (2021a). The idea is to use the network to predict $\hat{\boldsymbol{x}}_0$ from $\boldsymbol{x}_t$ and then use the forward model to move from $\hat{\boldsymbol{x}}_0$ to $\boldsymbol{x}_{t-\Delta t}$.

This is same with the Naive Sampler we introduced in the main paper but the difference in DDIM is that part of the stochasticity is replaced by "simulated" noise. The main trick to simulate the noise is to observe that once we know $\hat{x}_0$ and $x_t$, we can once again use the forward model to estimate the noise. We can use the same idea for Soft Diffusion sampling. Specifically, we propose the following DDIM-type sampler:

$$\boldsymbol{x}_{t-\Delta t} = \boldsymbol{C}_{t-\Delta t} \underbrace{\boldsymbol{h}_\theta(\boldsymbol{x}_t, t)}_{\hat{x}_0} + \sqrt{\sigma_{t-\Delta t}^2 - k^2} \underbrace{\frac{\boldsymbol{x}_t - \boldsymbol{C}_t \boldsymbol{h}_\theta(\boldsymbol{x}_t, t)}{\sigma_t}}_{\epsilon_\theta(x_t, t): \text{ simulated noise}} + k^2 \boldsymbol{\eta}. \qquad (52)$$

This Equation is very similar to Equation 12, page 5 in the DDIM (Song et al., 2021a) paper. The parameter $k$ controls the amount of stochasticity, i.e. how much of the noise is simulated. For $k = 0$, we get a deterministic sampler. Experiments with the deterministic DDIM-type sampler can be found in Section F.3.

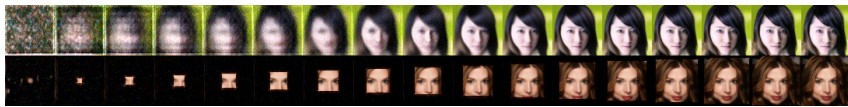

Figure 5: Conditional means $\mathbb{E}[\boldsymbol{x}_0 | \boldsymbol{x}_t]$ predictions of our blur/masking models, at different diffusion times.

## C  SCHEDULINGS

We use our framework to select the blur levels in an unsupervised way. For solving the optimal transport problem, we use the Sinkhorn Distances by Cuturi (2013) that has been extensively used for image experiments. We use the implementation of this paper from the software package `ott-jax` (Cuturi et al., 2022). We experimented with both using the whole dataset and using slices. As expected, dataset slices lead to higher estimation errors, but we observe that the found scheduling doesn't change much, i.e. the estimation error increases approximately uniformly for all the pairs for reasonably sized dataset slices. For the schedules found in the paper, we used slices of 10% of the dataset. We start with 256 different blur levels and we tune $\epsilon$ such that the shortest path contains 32 distributions. We then use linear interpolation to extend to the continuous case. Full experimental details can be found in the Appendix. These choices seem to work well in practice, but further optimization could be made in future work.

Figure 6 shows the found schedulings for the CelebA and the CIFAR-10 datasets. Notice that the scheduling is slightly different between the two datasets – the diffusion depends on the nature of the data we are trying to model. We underline that the parameters for the blur are selected without any supervision, by solving the optimization problem we defined in Section 3.3.

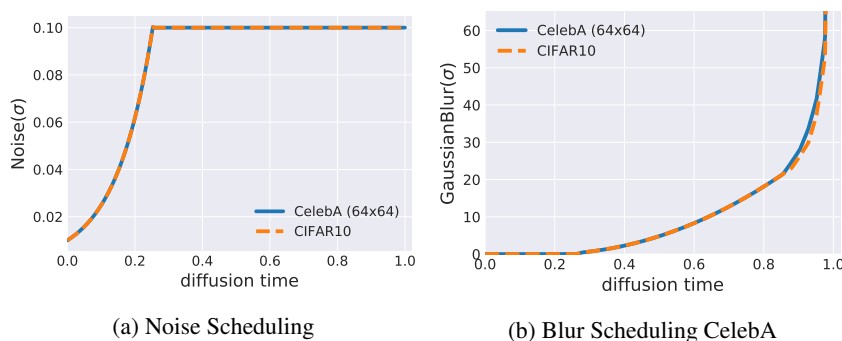

(a) Noise Scheduling          (b) Blur Scheduling CelebA

Figure 6: Diffusion scheduling for CelebA-64 and CIFAR-10. The blur corruption levels are selected without supervision to minimize the sum of the Wasserstein distances between consecutive distributions. Notice that the scheduling is slightly different between the two datasets – the diffusion depends on the nature of the data we are trying to model. The support of the Gaussian blur kernel was set to $65 \times 65$ and $161 \times 161$ for CIFAR-10 and CelebA-64 datasets respectively.

## D  TRAINING DETAILS

**Hyperparameters.**   For our trainings, we use Adam optimizer with learning rate $2e - 4$, $\beta_1 = 0.9$, $\beta_2 = 0.999, \epsilon = 1e - 8$. We additionally use gradient clipping for gradient norms bigger than $1$. For the learning rate scheduling, we use $5000$ steps of linear warmup. We use batch size $128$ and we train for $1 - 2M$ iterations (based on observed FID performance).

**Blur parameters.**   For the blurring operator, we use Gaussian blur with fixed kernel size and we vary the variance of the kernel. For CelebA-64, we keep the kernel half size fixed to $80$ and we vary the standard deviation from $0.01$ to $23$. For CIFAR-10, we keep the kernel half size fixed to $32$ and we vary the standard deviation from $0.01$ to $18$. For both datasets, we implement blur with zero-padding. We chose the final blur level such that the final distribution is easy to sample from. In both cases, the final distribution becomes noise on top of (almost) a single color.

**Final distribution**   For the blurring models, the final distribution is the distribution of very blurry images with additive Gaussian noise (we first blur, then add noise). At the limit, each blurry image becomes a constant image having a single color (i.e., the average color of the image). Hence, to sample from this final distribution, we first have to sample a single color (from the distribution of average colors in the dataset), generate a constant image having that color, and then add little noise to the constant image. The distribution of average colors for all the considered datasets is very simple and we model it with a Gaussian distribution. Specifically, we fit a three-dimensional gaussian distribution (one dimension for each color channel, diagonal covariance) to the average color of the dataset. To begin the inference process, we sample one color from this distribution, we create an image where every pixel has this color and then we add Gaussian noise.

**Architecture.**   We use the architecture of Song et al. (2021b) without any changes.

**Training Objective**   For all our experiments, we scale the loss at level $t$ with $w(t) = 1/\sigma_t^2$ as in Song & Ermon (2019; 2020); Song et al. (2021b).

**Compute and Training Time**   We train our models on 16 v2-TPUs. Our blur models on CelebA had an average speed of 6 iterations per second. For CIFAR-10, the average speed was 11 iterations per second We note that there is no overhead over the NCSN++ paper other than projecting to the measurements space, which can be done very efficiently for both blur and masking.

**Evaluation**   We keep one checkpoint every 10000 steps and we keep the best model among the kept checkpoints based on the obtained FID score. We use 50000 samples to evaluate the FID, as it is typically done in prior work. Regarding the number of steps, we selected the best FID among models evaluated with steps ranging from 200 to 1000 with a step size of 80 (10 experiments total).

The Momentum Sampler seems to have a U-shaped performance plot, i.e. there is a sweet spot in the number of function evaluations that gives the lowest FID score. We observed this in both CelebA and CIFAR-10.

However, this is not unique to our sampler – the general belief that sample quality improves with the number of steps has been called into question by recent works. Specifically, Karras et. al. do an extensive evaluation of the role of NFEs for different samplers and they find that many stochastic samplers behave similarly to our Momentum Sampler. For example, we refer the interested reader to Figure 4, page 8 of the paper Elucidating the Design Space of Diffusion-Based Generative Models that clearly shows that for many stochastic samplers performance slightly deteriorates after a certain threshold of function evaluations and the optimum is at some intermediate point. We emphasize, however, that the deterioration we observed in CelebA is bigger compared to other stochastic samplers and therefore selecting the correct number of NFEs is important for our method. Consistent with the observations by Karras et al. (2022) (e.g. Figure 2, page 4), for deterministic samplers the performance flattens (DDIM) or deteriorates slightly (Probability Flow Momentum Sampler) in the regime of very high Number of Function Evaluations (NFEs).

We also want to note that selecting the optimal number of function evaluations for reporting FID is not uncommon in the literature. It is done in both the landmark papers Elucidating the Design Space

of Diffusion-Based Generative Models (Karras et al., 2022) and Score-Based Generative Modeling through Stochastic Differential Equations (Song et al., 2021b).

## E    LIMITATIONS AND THINGS THAT DID NOT WORK

Our method has several limitations. First, it requires the diffusion operator to be known. This is not always the case, e.g. see Peebles et al. (2022) where diffusion is applied to *checkpoints* of different models. Another limitation is that our framework does not offer any guidance on *which* diffusion operators are actually more or less useful for learning the data distribution. Particularly, we already showed that blurring is a much more powerful diffusion mechanism than masking, in the sense that it leads to better FID scores and faster generations. It is yet to be seen whether blurring is going to be outperformed by some other corruption method. Our method also only concerns linear diffusions (however, the extension to non-linear is relatively straightforward).

On the theoretical side, our method has also some shortcomings. First, it only intuitively explains why the reparametrization to the Denoising Score Matching is needed. Second, since our method is based on the Denoising Score Matching, it only works when the conditional log-likelihood is defined everywhere. There are distributions for which such condition is not satisfied, but still, can be learned (to some extent) with heuristic methods (Bansal et al., 2022).

Another limitation of our work is the derivation of the Momentum Sampler and its sampling distribution. Consider the forward process:

$$\mathrm{d}\boldsymbol{x}_t = \dot{\boldsymbol{C}}_t \mathbb{E}_{\boldsymbol{x}_0 \sim p_0}[\boldsymbol{x}_0 | \boldsymbol{x}_t] + g(t)\mathrm{d}\boldsymbol{w} \tag{53}$$

This process is described by an Ito SDE, i.e. an SDE of the form:

$$\mathrm{d}\boldsymbol{x}_t = \boldsymbol{f}(\boldsymbol{x}_t, t)\mathrm{d}t + g(t)\mathrm{d}\boldsymbol{w}, \tag{54}$$

where $\boldsymbol{f}(\boldsymbol{x}_t, t) = \dot{\boldsymbol{C}}_t \mathbb{E}_{\boldsymbol{x}_0 \sim p_0}[\boldsymbol{x}_0 | \boldsymbol{x}_t]$.

The sampling process we write in the main body of the paper (that leads to the Momentum Sampler) is exactly the reverse of this forward process, where $\mathbb{E}_{\boldsymbol{x}_0 \sim p_0}[\boldsymbol{x}_0 | \boldsymbol{x}_t]$ is approximated by the neural network. Hence, the Fokker-Planck equations hold and we are guaranteed that (as long as the approximation of the conditional expectation and the approximation of the score function are accurate), we are sampling from the correct distribution (Song et al., 2021b).

During training, we cannot use this forward process because we do not have the conditional expectation. Instead, we replace the conditional expectation of $\boldsymbol{x}_0$ with $\boldsymbol{x}_0$ itself. The mismatch between $\boldsymbol{x}_0$ and the conditional expectation of $\boldsymbol{x}_0$ leads to an additional approximation error in the learning of the score. For small $t$, the conditional expectation will be very close to $\boldsymbol{x}_0$ and hence this approximation error is small. For large values of $t$, the distance between the conditional expectation and $\boldsymbol{x}_0$ is bigger. However, we are multiplying with the corruption matrix $\boldsymbol{C}_t$, which is removing information as $t$ grows. Therefore the distance of the corrupted conditional expectation and the corrupted $\boldsymbol{x}_0$ is also expected to be small. Our training process learns the correct score for the corruption process applied directly to $\boldsymbol{x}_0$. Our sampler, however, assumes that we have learned the score using the conditional expectation of $\boldsymbol{x}_0$. We intuitively expect that these two are not far away (for all $t$) as we explained. However, precisely characterizing this mismatch remains open.

On the practical side, we believe that our objective, Soft Score Matching, sometimes leads to slower sampling compared to the simpler objective of predicting the clean image. For example, for masking, since the model is only penalized in the observed region, there is no incentive in expanding this region. Hence, to achieve smooth transition between different masking levels we need to run many steps.

Experimentally, we tried using our framework with even less stochasticity, but it did not work, e.g. see 7. It would be interesting to understand better what is causing the failure and also what is the proper amount of randomness required at each diffusion step.

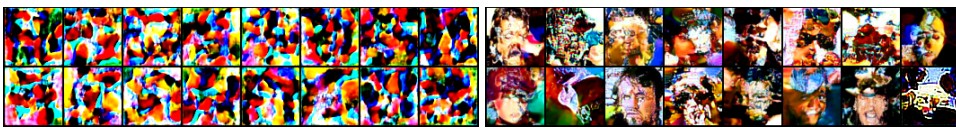

(a) Images generated with model trained with $\sigma_{\max} = 0.025$

(b) Images generated with model trained with $\sigma_{\max} = 0.05$

Figure 7: Ablation study for the magnitude of noise.

## F    ADDITIONAL RESULTS

### F.1    SCHEDULING ABLATIONS

#### F.1.1    ABLATION STUDIES FOR NOISE

In the experiments of the main paper, our diffusion involves an initial stage where only noise is added. Then noise is fixed and the images are getting corrupted by the deterministic operator (e.g. blur or masking). In this section, we show two ablations regarding the noise.

**Magnitude of noise.**    In this ablation study, we still keep the noise fixed for a significant part of the diffusion, but we ablate the magnitude of the noise. Specifically, we attempt to study to what extent noise is needed in order to learn to reverse corruption processes with Soft Score Matching. We train two additional models on CelebA-64 where we attempt to decrease the maximum noise to a lower value. The corruptions for both models involves an initial stage where noise grows geometrically rate from the initial value (0.01) to the maximum value. Then, all the models keep the noise fixed at their maximum value for the rest of the diffusion. We use the following maximum values: i) $\sigma_{\max} = 0.1$ (model used in the paper), ii) $\sigma_{\max} = 0.05$, and iii) $\sigma_{\max} = 0.025$. Unconditional samples from the two ablation models are shown in Figure 7. As shown, both models fail to produce realistic samples. The quality of samples deteriorates significantly as the noise decreases – the samples from the ablation models are significantly worse than the ones produced by our state-of-the-art model (see Figure 2 (right)). We want to underline that this is not a conclusive study. It might be the case that with different hyperparameters one can make Soft Score Matching work with lower values of noise. For example, we might need to tune the weights $w(t)$ since for the ablations (and the state-of-the-art model), we use $w(t) = 1/\sigma_t^2$ (as in Song & Ermon (2019; 2020); Song et al. (2021b)) which might be causing instabilities for low values of noise (Nichol & Dhariwal, 2021).

**Noise changing throughout the diffusion.**    We also train a model where noise and blur are changing simultaneously throughout the diffusion. This is a sanity check to verify that our framework (learning and sampling) still works when the model needs to deblur and denoise at the same time. For the noise scheduling, we simply use geometric scheduling from 0.1 to 0.01. We keep the blur parameters the same with the state-of-the-art model, i.e. we use the blur parameters shown in Figure 6. This model achieves a competitive FID score, 4.31. This score could probably be further improved (by jointly selecting the blur and the noise scheduling with our framework), but this is beyond the scope of this ablation.

#### F.1.2    ABLATION STUDY FOR BLUR

For all our experiments so far, we chose the blur corruption levels based on the scheduling method we described in Section 3.3. We show the benefits of our approach by comparing to a natural baseline for selecting the diffusion levels. For this natural baseline, we use the scheduling of Variance Exploding (VE) as guidance. Specifically, we choose the blur parameters such that the MSE between the corrupted image and the clean image decays with the same rate for the Gaussian Denoising Diffusion and our Soft Diffusion (blur and low magnitude noise). Formally, let $\{q_t'\}_{t=0}^1$ be the (noisy) distributions used in Song et al. (2021b) for the Variance Exploding (VE) SDE and let $\{q_t\}_{t=0}^1$ the blurry (and noisy) distributions we want to select. At level $t$, we choose the blur parameters such

that:

$$\frac{\mathbb{E}_{(\boldsymbol{x}_0, \boldsymbol{x}_t) \sim q_t(\boldsymbol{x}_t | \boldsymbol{x}_0) q_0(\boldsymbol{x}_0)} \left[ ||\boldsymbol{x}_0 - \boldsymbol{x}_t||^2 \right]}{\mathbb{E}_{(\boldsymbol{x}_0, \boldsymbol{x}_1) \sim q_1(\boldsymbol{x}_1 | \boldsymbol{x}_0) q_0(\boldsymbol{x}_0)} \left[ ||\boldsymbol{x}_0 - \boldsymbol{x}_1||^2 \right]} = \frac{\mathbb{E}_{(\boldsymbol{x}_0, \boldsymbol{x}_t) \sim q_t'(\boldsymbol{x}_t | \boldsymbol{x}_0) q_0(\boldsymbol{x}_0)} \left[ ||\boldsymbol{x}_0 - \boldsymbol{x}_t||^2 \right]}{\mathbb{E}_{(\boldsymbol{x}_0, \boldsymbol{x}_1) \sim q_1'(\boldsymbol{x}_1 | \boldsymbol{x}_0) q_0(\boldsymbol{x}_0)} \left[ ||\boldsymbol{x}_0 - \boldsymbol{x}_1||^2 \right]}. \quad (55)$$

We retrain on CelebA using this natural baseline method for selecting the diffusion parameters. For a fair comparison, we keep the architecture and all the hyperparameters the same and we only ablate the scheduling of the blur. *We measure FID for both the trained model with the baseline scheduling and we observe it increases from* $1.85$ *to* $8.35$. Apart from this large deterioration in performance, the baseline model obtains its best FID score after 2000 steps, while with our scheduling we only need 280 steps to obtain the best FID. This experiment shows that the choice of scheduling is really important for the model performance but for also the computational requirements of the sampling.

### F.2 CIFAR-10: ADDITIONAL RESULTS

We report how FID is changing based on the Number of Function Evaluations for CIFAR-10 with Momentum Sampling in Figure 8. The baseline model we are comparing against is NCSN++ trained with VE SDE, since our model also uses a VE SDE. For the baseline, we use the Reverse SDE sampler with Euler–Maruyama discretization (similar to our Momentum Sampler). The numbers are taken directly from Ma et al. (2022).

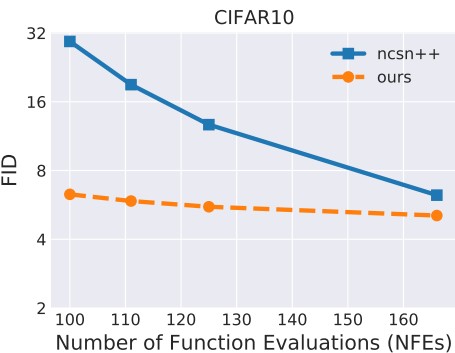

Figure 8: Demonstration of how FID changes based on Number of Function Evaluations (NFEs) for CIFAR-10 for our (blur) model on CIFAR-10 with momentum sampling. Our model offers significant performance benefits for low number of function evaluations.

We also show visual samples from our blurring model, NCSN++ (VE SDE) and DDPM++ (VP SDE) in Figure 9. For all the models we fix the NFEs to 200 and we show samples generated with the Euler-Maruyama discretization of the associated reverse ODE. As shown, the baseline models generate samples with artifacts for low NFEs while our model leads to images of superior visual quality. We note that different samplers can be used to accelerate sampling for all models, e.g. samplers from Karras et al. (2022) or DDIM-type samplers (Song et al., 2021a), as we show in Section B.2.

We also report best FID performance for different samplers for ours and competing methods in Table 2. Our best FID is competitive with similar samplers to similar methods, e.g. with NCSN++ trained with VE SDE and using the Reverse SDE sampler. However, it is significantly behind the state-of-the-art LSGM (Vahdat et al., 2021) model. We believe that this performance gap can be decreased in the future by further research in the area of diffusion models with general corruptions.

### F.3 SAMPLING ABLATIONS

We perform several ablations regarding the sampling method, additional to the results mentioned in the paper.

**DDIM** We experiment with the deterministic DDIM-type sampler and we report our results in the Table 3 for CelebA-64. Our DDIM-type sampler is very effective when the number of function evaluations is very low – it achieves FID $5.08$ with only $50$ steps while in comparison, the Momentum

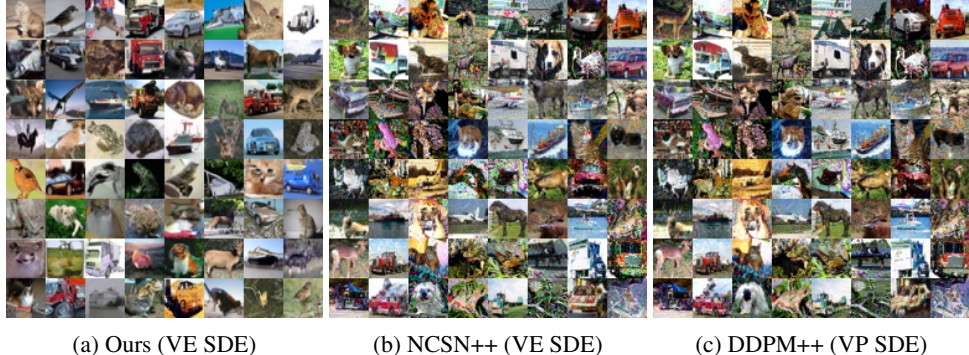

(a) Ours (VE SDE)         (b) NCSN++ (VE SDE)        (c) DDPM++ (VP SDE)

Figure 9: Visual comparison of samples from our model and baselines for 200 NFEs. For all the models, we are using the Euler-Maruyama discretization of the associated reverse ODE to generate the samples. As shown, our model leads to superior visual quality.

| Method | FID |
|---|---|
| Ours (VE SDE) Naive Sampler | 40.07 |
| Ours (VE SDE) Momentum Sampler | 3.91 |
| Ours (VE SDE) Probability Flow Momentum Sampler | 3.86 |
| NCSN++ (VE SDE) Reverse SDE | 4.79 |
| NCSN++ (VE SDE) Probability Flow | 10.54 |
| DDPM (VP SDE) | 3.17 |
| LSGM (Vahdat et al., 2021) | **2.10** |

Table 2: FID Results on CIFAR-10 for different samplers. For each of our samplers, we report the best result obtained among 10 runs with different NFEs, ranging from 200 to 1000 with a step size of 80. The results for competing methods are taken directly from the papers.

Sampler, achieves FID 30.31 with 50 steps. The efficiency of DDIM is consistent with what has been observed in (Song et al., 2021a; Karras et al., 2022). DDIM's performance is also not getting worse as we increase the number of steps. Instead, the performance of the Momentum Sampler, as observed in Figure 3 has a U-shape: there is an intermediate optimal number of steps to achieve the best FID score. We believe this could be related to the fact that Momentum Sampler is not deterministic. Specifically, in Karras et al. (2022) it is observed that for deterministic samplers performance usually flattens after a point (e.g. see Figure 2, page 4) while for some stochastic samplers performance goes again up after a certain number of function evaluations (e.g. see Figure 4, page 8).

| NFEs | FID |
|---|---|
| 25 | 10.54 |
| 50 | 5.08 |
| 100 | 3.40 |
| 200 | 2.92 |
| 300 | 2.83 |
| 600 | 2.80 |
| 1000 | 2.79 |

Table 3: Results of DDIM (Song et al., 2021a) sampler for CelebA-64 dataset using our blurring model. The DDIM sampler is particularly effective for low number of function evaluations and maintains its performance as we increase the number of steps.

**Probability Flow Momentum Sampler**    For CelebA-64, we reported results only for the (stochastic) Momentum Sampler in the main body of the paper (see Figure 3). We report here results for the deterministic counterpart of this sampler, Probability Flow Momentum Sampler, summarized in Table 4. The deterministic version of the Momentum Sampler seems to be performing slightly

worse than the stochastic version – FID jumps from 1.85 to 2.14. However, this sampler, similar to the DDIM-type sampler, maintains more performance as we increase the number of steps. This is an important advantage for the deterministic sampling methods since practitioners need to put less effort into tuning the NFEs to get the best result.

| NFE | FID |
|---|---|
| 50 | 13.77 |
| 100 | 3.27 |
| 200 | 2.14 |
| 300 | 2.21 |
| 600 | 2.59 |
| 1000 | 2.49 |

Table 4: Results of the Probability Flow Momentum Sampler for Celeba-64 for our blurring models.

**Predictor-Corrector Samplers** Finally, we perform experiments with Predictor-Corrector samplers, as proposed in Song et al. (2021b). The idea is that we are alternating at each diffusion step between two different samplers. We experiment with a DDIM Predictor and a Probability Flow Momentum Sampler corrector. The results are summarized in Table 5. The Predictor-Corrector sampler maintains some of the benefits of both samplers. Namely, for low number of function evaluations it has better performance than the Probability Flow Momentum Sampler (benefit coming from the DDIM sampler) and for higher number of function evaluations performance is better than the DDIM sampler (benefit coming from the Probability Flow Momentum Sampler). There is a spot, at 100 NFEs, where the Predictor-Corrector sampler is better than both the Predictor and the Corrector. We encourage future research in identifying even better pairs of Predictor-Correctors that might outperform both the Predictor and the Corrector in some regime.

| NFE | FID |
|---|---|
| 25 | 10.56 |
| 50 | 5.21 |
| 100 | 3.21 |
| 200 | 2.86 |
| 300 | 2.71 |
| 600 | 2.73 |
| 1000 | 2.73 |

Table 5: Results of a Predictor-Corrector (Song et al., 2021b) sampler for Celeba-64 for our blurring models. The Predictor is the DDIM-type sampler and the corrector our Probability Flow Momentum Sampler.

### F.4    NEAREST SAMPLES IN TRAINING DATA

To verify that our model does not simply memorize the training dataset, we present generated images from our model and their nearest neighbor (L2 pixel distance) from the dataset. The results are shown in Figure 10.

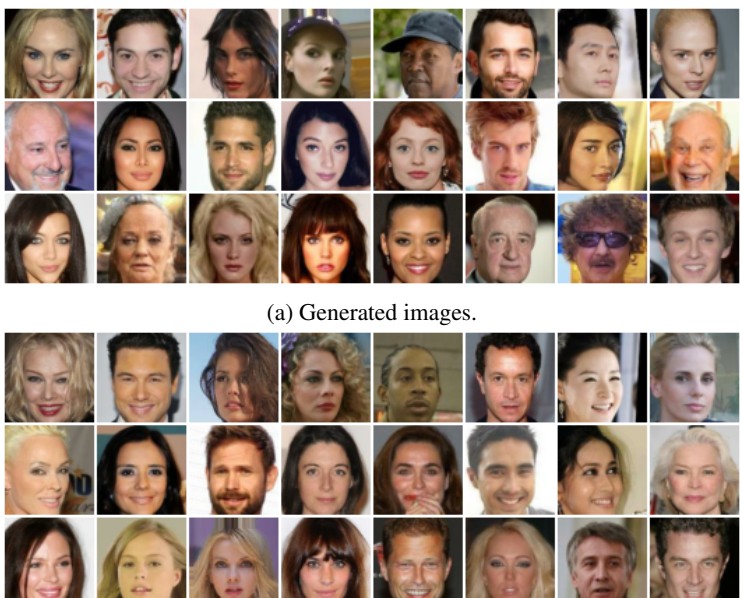

(a) Generated images.

(b) Nearest neighbors from dataset.

Figure 10: Generated images and nearest neighbors (L2 pixel distance) from the training dataset. As shown, the model produces new samples and does not simply memorize the training dataset.

## F.5 UNCURATED SAMPLES

Figures 11 and 12 show more uncurated samples from our trained models on CelebA and CIFAR-10.

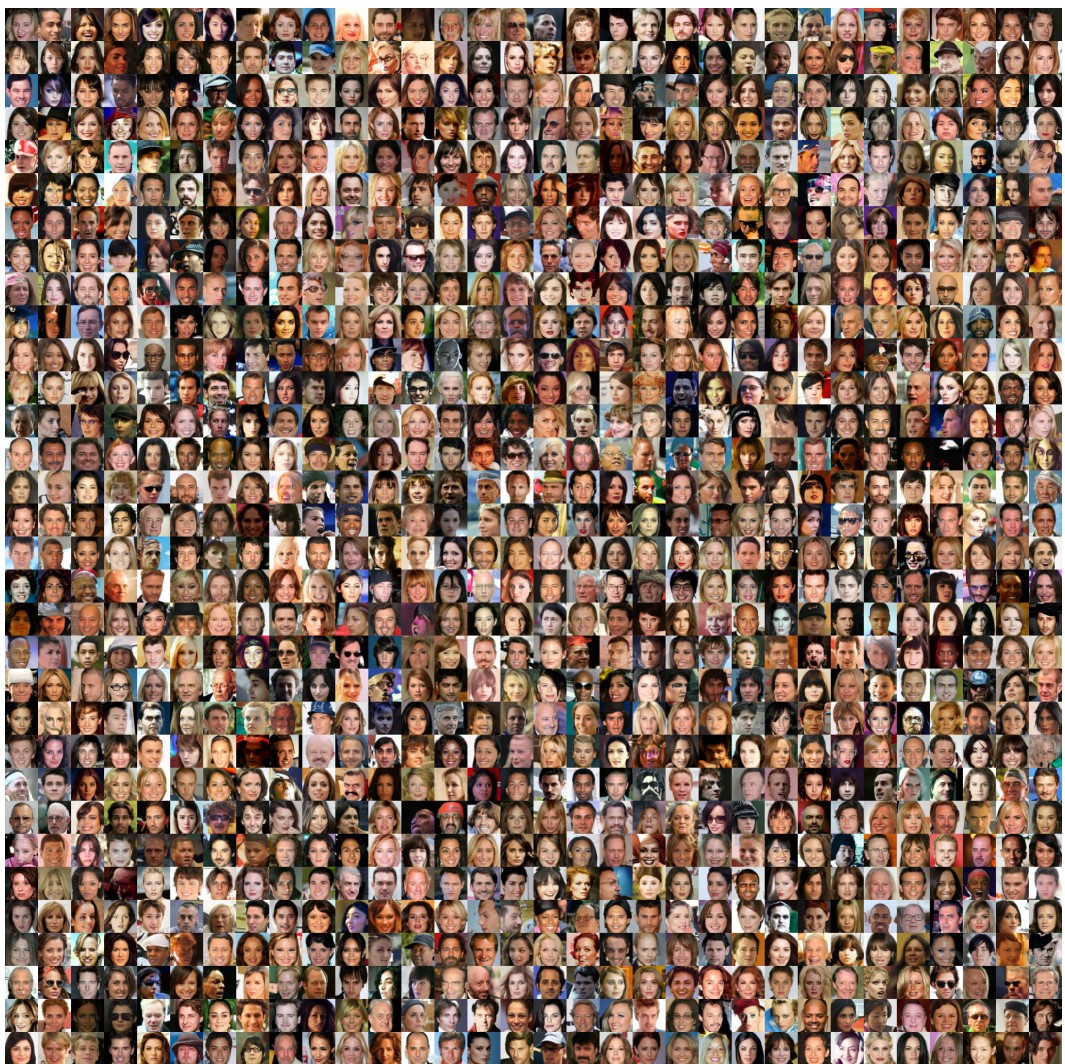

Figure 11: More uncurated samples from our blur model trained on CelebA. These samples are obtained using our Momentum Sampler with 280 NFEs.

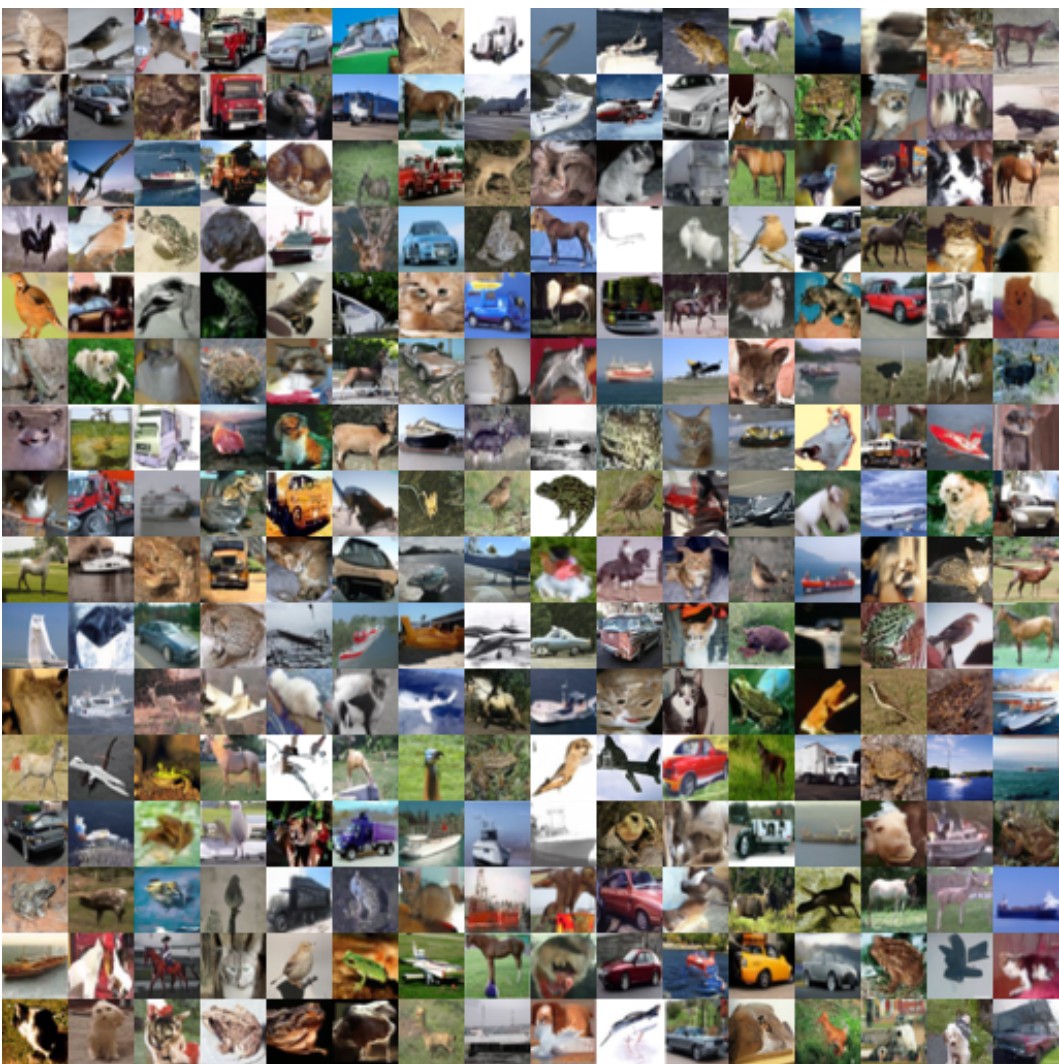

Figure 12: More uncurated samples from our blur model trained on CIFAR-10. These samples are obtained using the Probabilistic Flow Momentum Sampler with 200 NFEs.

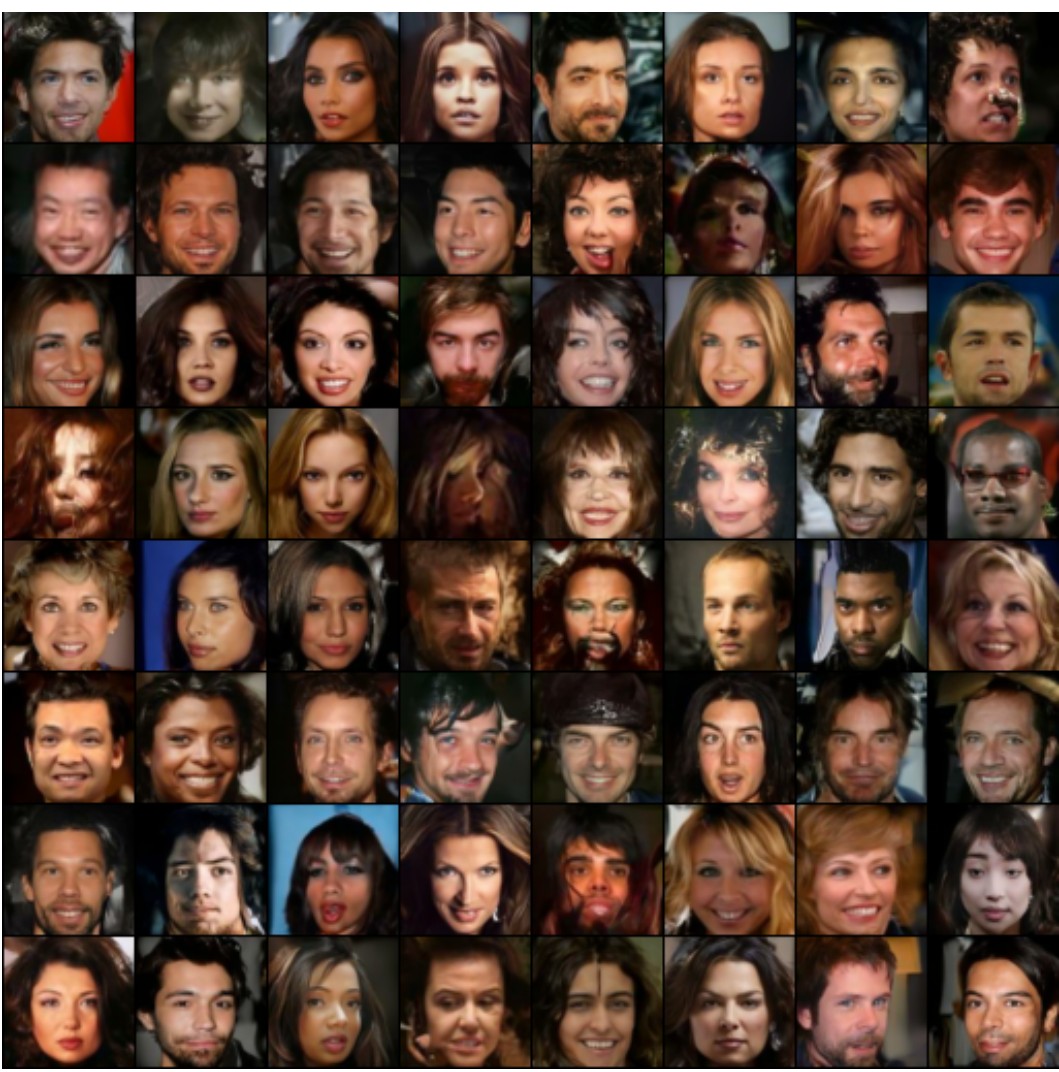

Figure 13: Uncurated samples from our masking model trained on CelebA. These samples are obtained using the Probabilistic Flow Momentum Sampler with 1024 NFEs.

## G   IMPLEMENTATION OF DEGRADATION OPERATORS

**Blurring.**   The blurring operator is implemented as a convolution with a Gaussian kernel. The Gaussian kernel is truncated to have fixed support size ($161 \times 161$ pixels), and normalized to have area one.   The standard deviation of the Gaussian kernel `sigma_blur` (parameter controlling the strength of the blur) is computed using the optimal transport optimization for the scheduling.   The obtained schedule for the parameter `sigma_blur` in CIFAR10 can be accessed from the following anonymous url: `https://drive.google.com/file/d/192kdbj9oq1EGCm7KY52QZj--x0g5Elgs`.

```
def generate_gaussian_kernel(sigma_blur, half_size=80):
  v = np.arange(-half_size, half_size + 1)
  x, y = np.meshgrid(v, v)
  k = np.exp(-(x**2 + y**2) / (2*sigma_blur**2))
  k = k / np.sum(k)
  return k
```

**Masking.**   The masking operator is implemented using a centered binary mask that sets to zero some percentage of pixels in the image.   The scheduling with the masking operators can be accessed from the following anonymous url: `https://drive.google.com/file/d/1YjzYKgivhvbHOzTABOYDiUoHbzq60Ozy`.

**Noise.**   The noise degradation is implemented by adding to the blurred or masked image white Gaussian noise having standard deviation `sigma_noise`. The scheduling of the noise can be accessed from the following anonymous url: `https://drive.google.com/file/d/17UwFlJyp4euQeKVAVmRRHg8HL52anapf`.

