# OpenReview forum: "Soft Diffusion: Score Matching For General Corruptions"
_ICLR.cc/2023/Conference — Submitted to ICLR 2023_

### Official Review · Reviewer_x7zn · 2022-10-25

**Confidence:** 4
**Correctness:** 2
**Technical Novelty And Significance:** 3
**Empirical Novelty And Significance:** 3
**Recommendation:** 3

**Clarity, Quality, Novelty And Reproducibility:**

-

**Strength And Weaknesses:**


Strength:

(1) Build a framework for diffusion with general linear corruptions.

(2) Corruption scheduling using the Wasserstein metric seems interesting.

(3) Momentum sampler seems interesting.



Weakness:

(1) The authors should provide specific forms of $C_t$ for Gaussian blur and masking.

(2) The scheduling of the diffusion is unclear. e.g., how do you estimate the Wasserstein distance? How is the noise level $s_t$ scheduled? (Figure 6 alone is not enough for replicating the result.) What is the tunable parameter in $C_t$?

(3) Experiments: you should faithfully report your results on CIFAR10, and summarize them as a Table.

(4) To argue about the superiority of the sampling speed, the authors should early-stop the sampling process, e.g. at the 200 steps, and provide a visualization of examples.

(5) About Figure 3: Why only compare your method with NCSN++? what about DDPM++, which is the previous SOTA as reported in Table 1?

**Summary Of The Paper:**

The paper extends the original diffusion models to linear corruptions, which include Gaussian blur and masking. Motivated by the SDE form of the backward process, the paper proposes the so-called "momentum sampler".
The noise/corruption scheduling are selected based on some Wasserstein matric. Empirically, they improve the FID score from previous SOTA (1.90) to 1.85 on CelebA 64x64.

**Summary Of The Review:**

The delivery of the paper should be improved. The experimental sections should be improved.

---

> ### Comment · Program_Chairs · 2022-11-10
> **no concerns of plagiarism**
>
> PC team confirmed with the authors of [1] (which the above reviewer raised concerns) that this submission is "both different enough and simultaneous enough" that they should be "considered as independent contributions" and that the submission "cites my paper and clearly explains the differences". Given this clear evidence, PC team concluded that there is no ethical concern of plagiarism of this submission. We thank both authors of [1] and authors of this submission for clarifying this point.
>
> -SPC

---

> > ### Comment · Reviewer_x7zn · 2022-11-16
> > **Thank you**
> >
> > I sincerely thank the SPC, PC team, the authors of this paper, and the authors of [1] for the joint effort to erase my ethical concern.

---

> ### Author Response · Authors · 2022-11-11
> **Response to Reviewer x7zn**
>
> **Cold Diffusion and other concurrent works**
>
> We have reached out to the Program Chairs, who after consultation with authors of [1], have confirmed that the accusation of plagiarism is entirely baseless. We encourage the reviewer to amend their statements and refrain	from making further defamatory remarks.
>
> The **concurrent** work “Cold Diffusion” is cited and discussed **six times** in our submission. We have presented a detailed discussion and comparisons to it in our submission. The authors of [1] have also confirmed that our work is both independent and concurrent. The Cold Diffusion work is an excellent and inspirational paper that shared a similar goal as ours. It is a concurrent work, to which we have assigned proper credit and reference in our submission.
>
> Our work and Cold Diffusion are just some of several concurrent works to employ various types of corruption. Other examples submitted to ICLR include the following:
>
> * [Restoration Based Generative models](https://openreview.net/pdf?id=iNUtsk4h2q1), ICLR submission.
> * [Blurring Diffusion Models](https://openreview.net/forum?id=OjDkC57x5sz), ICLR submission.
> * [Generative Modelling With Inverse Heat Dissipation](https://openreview.net/forum?id=4PJUBT9f2Ol), ICLR submission.
> * [Diffusion models in space and time via the discretized heat equation](https://openreview.net/forum?id=BrlGyp4uDbc), Submitted to ICLR Workshop on Deep Generative Models for Highly Structured Data, 2022.
>
> All of the above works use blurring as a corruption mechanism; and works that appeared before are properly cited and discussed in our manuscript. As such, we again encourage the reviewer to reconsider their remarks, and kindly refrain from dispensing terms like “scooped” in a way that seems intended more to denigrate the authors rather than engage in a helpful debate.
>
> **Technical comparisons to Cold Diffusion.**
>
> In Soft Diffusion we take a very different approach for generalizing diffusion compared to other concurrent works, and in particular to Cold Diffusion. We provide a new framework for (i) training, (ii) scheduling and (iii) sampling.
>
> One of the major differences with respect to Cold Diffusion is that their approach explicitly avoids the use of noise. Meanwhile in Soft Diffusion, we generalize diffusion to other degradations, but we always require noise, as it plays a critical role to satisfy the technical requirements in our score-matching formulation.
>
> The reviewer claims that we are using the same loss as in Cold Diffusion. We respectfully disagree.
>
> We are using a probabilistic score-matching loss, whereas Cold Diffusion is directly predicting the clean image from an arbitrary corruption.
> Our loss is on the filtered space, while Cold Diffusion loss is on the clean image domain.
>
> There is a clearly marked paragraph "Comparisons with objectives used in other works" in Section 3 (page 4), that delineates the differences. There is also extended discussion in the Related Work Section (page 9).
>
> The reviewer asks for a detailed comparison with Cold Diffusion [1]. Such a comparison has already been given. Specifically, we refer the reviewer to the “Training Objective” paragraph in the Experimental Section (page 7) where we compare with the Cold Diffusion loss.
>
> With identical corruption and hyperparameters, the FID score worsens (from 1.85 to 5.91) if we use the loss in [1]. Also, [1] reports a FID 80.08 on CIFAR-10, whereas our approach yields FID of 3.86.
> The reviewer claims that the introduced Momentum Sampler is the same as in Cold Diffusion. This is untrue. To re-iterate, noise plays a major role in our formulation. Our sampler is specific to the case where noise is introduced. If one limits the sampler to the case where there is no noise, the special case coincides with the one introduced in Cold Diffusion. Put differently, our update rule given in Eq. (16) can be interpreted as a generalization of the sampler used in Cold Diffusion; and it is obtained with completely different techniques – namely, by looking at the Forward SDE. We refer the reviewer to Section F.1.1 in the Appendix and particularly to Figure 7.

---

> > ### Comment · Reviewer_x7zn · 2022-11-16
> > **Thank you for your additional clarification**
> >
> > Dear authors,
> >
> > First, I would like to make clear that as a reviewer, I am obliged to raise concerns on any possible potential violations of the ICLR Code of Ethics; nevertheless, such a concern is not any accusation by default, and it requires public scrutiny to be a consensus. I do not intend to make any defamatory assertion, neither in the past nor in the future.
> >
> > Next, I would like to thank the authors again for the additional clarifications on the differences between your paper and [1]. I give you my sincerest apology that I have indeed missed any of the discussions of your work with [1]. I have removed the flag for ethnic review and my concerns about technical and empirical novelties in my original review and changed the scores accordingly. Still, I shall retain my opinions on the similarities among the essence of the series of papers.
> >
> >
> > [1] Cold Diffusion: Inverting Arbitrary Image Transforms Without Noise. Arpit Bansal, Eitan Borgnia, Hong-Min Chu, Jie S. Li, Hamid Kazemi, Furong Huang, Micah Goldblum, Jonas Geiping, Tom Goldstein

---

> > > ### Author Response · Authors · 2022-11-17
> > > **Further Clarifications**
> > >
> > > We thank the Reviewer for the acknowledgement, we appreciate the removal of concerns of plagiarism and this public acknowledgement.  The Reviewer's current recommendation, however, remains as ‘3: reject, not good enough’. The Reviewer clearly states that this recommendation is because: **"Still, I shall retain my opinions on the similarities among the essence of the series of papers.’"** Similarities to concurrent and independent work cannot be grounds for recommending rejection, in our opinion. We ask the Reviewer to please consider updating the final recommendation.
> > >
> > > Further clarifications are given below.
> > >
> > > **"The authors should provide specific forms of C_t for Gaussian blur and masking.”**
> > >
> > > We thank the reviewer for their suggestion. We added in Section G of the Appendix pseudocode and pointers (through anonymous urls) to files with the parameters to generate the blur and masking operators.
> > >
> > >
> > > **"The scheduling of the diffusion is unclear. e.g., how do you estimate the Wasserstein distance? How is the noise level scheduled? (Figure 6 alone is not enough for replicating the result.) What is the tunable parameter in C_t?”**
> > >
> > > We use the `ott-jax` software package to estimate the pairwise Wasserstein distances. The software package implements the Sinkhorn Distances from the NeurIPS 2013 paper Sinkhorn Distances: Lightspeed Computation of Optimal Transport. This estimation algorithm is very fast which is very convenient for scaling to image datasets. We also refer the Reviewer to our response to Reviewer pNiX and the details we provide in Section C (Schedulings) of the Appendix – page 16.
> > >
> > > Regarding the noise level scheduling: as we explain in Section 4, page 7 of the main body of the paper, we use the typical geometric scheduling for the Variance Exploding SDE to increase the noise value to a certain threshold and then we keep it constant (for the vast majority of the diffusion time, see also Figure 6). There are further ablations regarding the scheduling of the noise in the Appendix, Section F.1, page 17. The noise parameters can also be found in [here](https://drive.google.com/file/d/17UwFlJyp4euQeKVAVmRRHg8HL52anapf).
> > >
> > > Regarding the tunable parameter in $C_t$: For the blurring parameters, we keep the Gaussian blur kernel size fixed to a large value (161x161) and we tune the standard deviation of the Gaussian kernel. For the masking experiments, the tunable parameter is the number of the observed pixels – for our masking experiments we use a mask growing from the center to the edges, as shown in Figure 1. For non-integer sizes we just linearly interpolate among the two neighbors. Please see the newly introduced Section G in Appendix.
> > > We are happy to answer further questions if needed.
> > >
> > > **“Experiments: you should faithfully report your results on CIFAR10, and summarize them as a Table.”**
> > >
> > > We thank the Reviewer for the suggestion. The results for CIFAR-10 are now summarized in Table 2.
> > >
> > > **“To argue about the superiority of the sampling speed, the authors should early-stop the sampling process, e.g. at the 200 steps, and provide a visualization of examples.”**
> > >
> > > We thank the Reviewer for the suggestion. **We added Figure 9**, in which we compare visually: i) our method, ii) NCSN++ (VE SDE), iii) DDPM++ (VP SDE) for 200 NFEs. For all models, we use the first-order discretization of the Reverse SDE for sampling. As shown, our method clearly outperforms the baselines -- the competing methods give samples with artifacts for low NFEs. We want to clarify that it is possible to improve the sampling speed of all methods (including ours) by using newer samples, e.g. in Table 3 we show significant improvements for low NFEs using DDIM. However, extensively exploring all the different sampling methods is out of the scope of this paper.
> > >
> > > **"About Figure 3: Why only compare your method with NCSN++? what about DDPM++, which is the previous SOTA as reported in Table 1?"**
> > >
> > > We compare it with the NCSN++ because it is also using a VE SDE and is based on the first-order discretization of the Reverse SDE, as in our Momentum Sampler. We also added a visual comparison with DDPM++ in Figure 9 (see also response to the previous point of the Reviewer). We clarify in the paper (Related Work Section) that there is a huge line of recent work on accelerating sampling. All these methods could be independently combined with our framework to get further accelerations. For example, we now **added** in the Appendix **experiments with DDIM-type samplers** (Table 3) and we got significant improvements for low NFEs. We leave experimenting with other recent samplers for future work.

---

### Official Review · Reviewer_pNiX · 2022-10-25

**Confidence:** 4
**Correctness:** 2
**Technical Novelty And Significance:** 2
**Empirical Novelty And Significance:** 2
**Recommendation:** 3

**Clarity, Quality, Novelty And Reproducibility:**

Clarity: good
Quality: good
novelty: marginally
Reproducibility: looks good

**Strength And Weaknesses:**

Strength:

1. Authors show denoising score matching~(DSM) can be used to learn score for noised diffusion with general linear corruption processes theoretically.

2. With the proposed several training and sampling improvements, diffusion models with general linear corruption can achieve competitive sampling quality on CIFAR10 and CelebA.

Weaknesses:

**general corruption**

1. Theoretically, extending diffusion models from the isotropic corruption process to general non-isotropic corruption is quite straightforward based on original DSM work or recent.

2. Regarding performance, the claimed SOTA FID on CelebA 64 is not accurate,  where [3,4]  have lower FID.

3. The performance of the proposed diffusion model in popular CIFAR-10 is worse than VPSDE or its variants. Authors compare with a weak baseline VESDE with reverse diffusion and ODE solver, whose performance is much better than the suggested Predictor-Corrector sampler. Authors claim "it is known[5] that diffusion models with VP SDEs usually do better in CIFAR-10". However, [5] suggests their VESDE can beat previous VPSDE and VESDE, and different diffusion models with isotropic corruption can be transformed into VESDE. The performance difference is due to various pre-condition and noise schedules during training.

4. To the best of the reviewer's knowledge, the SDE of the general corruption, which has a general linear shift term, can also be transformed into VESDE with a similar technique in [5] or exponential transformation introduced in [6].

**Momentum Sampler**

1. The idea is quite interesting. It will be nice to compare with SOTA sampler in diffusion models, such as ODE solvers introduced in [5,6].

**Scheduling**

1. The idea that choosing scheduling based on minimal distance from D0 to D1 is new. However, there is no theoretical evidence to justify why minimizing distance with the selected w2 distance is better. It is hard to predict the benefits of the choice or show the choice can minimize discretization error when solving sampling SDE or ODE.

2. Besides, estimating Wasserstein distances accurately between two datasets is quite expensive, even in CIFAR10 32x32 datasets. It would be nice to include more details, such as whether authors use partial datasets or some approximations in solving optimal transportation problems in the image dataset scale.



[3] Kim et al. Maximum Likelihood Training of Implicit Nonlinear Diffusion Model
[4] Wang et al. Diffusion-GAN: Training GANs with Diffusion
[5] Karras et al. Elucidating the Design Space of Diffusion-Based Generative Models
[6] Zhang et al. Fast Sampling of Diffusion Models with Exponential Integrator

**Summary Of The Paper:**

The work explores general linear corruption processes for diffusion models and practical training and sampling to improve performance for the new diffusion models. However, there are several technical questions unclear.

**Summary Of The Review:**

In summary I think the proposed method is interesting and technically sound. But the benefits over previous methods are not clear both in theory and practice. However, the benefits over previous methods are not clear both in theory and practice. I would like to increase my score if the authors can address my questions.

---

> ### Author Response · Authors · 2022-11-11
> **Response to Reviewer pNiX: Part 1, Summary of Revisions**
>
> We thank the Reviewer for their valuable comments. We are glad that the Reviewer finds our method interesting and technically sound. We submitted a new revision of our paper that addresses the concerns of the Reviewer. Below, we summarize the key points of our response for visibility. All the points are addressed in detail in the next comments.
>
> **Summary of changes**
>
> * We experiment with more samplers, as the Reviewer recommended. We show how we can use DDIM-type samplers in our framework. Our new DDIM samplers (presented in Section B2 of the Appendix, results in Table 3) significantly outperform our Momentum Sampler and our Probability Flow Momentum Sampler in the regime of low number of function evaluations. FID improves from $30.31$ (Momentum Sampler) for $50$ NFEs to $5.08$ with our new DDIM samplers. For higher NFEs, DDIM-type samplers perform worse than our samplers with momentum. We thank the Reviewer for this valuable suggestion. We also experiment with Predictor-Corrector samplers and we show that in certain regimes they are better than both the Predictors and the Correctors. Finally, we add more experiments with the Probability Flow ODE.
> * We clarify that, by the time of the submission, our method achieves state-of-the-art on CelebA among linear diffusion models. Among all the available models, to the best of our knowledge, our method ranks 2nd. The paper [Maximum Likelihood Training of Implicit Nonlinear Diffusion Models
> ](https://arxiv.org/abs/2205.13699) slightly improves our FID from $1.85$ to $1.75$ by using non-linear diffusion with normalizing flows. The revision of the [Diffusion-GAN](https://arxiv.org/abs/2206.02262) paper that further boosts the FID score was made available **after the ICLR deadline**.
> * We explain in detail the difference of our work with other works that consider linear drifts. In short, our corruption process is not Markovian and hence it cannot be described by an Ito SDE. The novelty of the Momentum Sampler is *approximating* the corruption process with a suitable Ito SDE (that can later be solved with any of the methods in the literature).
> * We further explain the differences of Soft Score Matching with previously proposed objectives in the literature (Cold Diffusion, DSM).
> * We provide further details on the computation of Wasserstein distances that are needed for the proposed unsupervised scheduling algorithm.
>
>
> We thank the Reviewer for the useful feedback and for their willingness to raise the score if the concerns are addressed. The aforementioned points are discussed analytically below. If there are further clarifications needed, we would be happy to address them during the rebuttal period.

---

> > ### Author Response · Authors · 2022-11-11
> > **Response to Reviewer pNiX: Part 2, Experimental Evaluation**
> >
> > In this response, we address questions of the Reviewer about our experimental evaluation.
> >
> > **"Regarding performance, the claimed SOTA FID on CelebA 64 is not accurate, where [3,4] have lower FID."**
> >
> > We first want to mention that the result on CelebA for [4] was made public after the ICLR deadline. Submissions [v1](https://arxiv.org/pdf/2206.02262v1.pdf), [v2](https://arxiv.org/pdf/2206.02262v2.pdf) on arXiv that were made public on June 5 and June 17, do not have results on CelebA. The submission that adds the state-of-the-art result on CelebA is made on Oct 9, 2022 which is after the ICLR deadline.
> >
> > In the paper, we claim state-of-the-art for linear diffusion models. E.g. in the abstract we mention: “outperforming all previous linear diffusion models” and later on we state “Our trained models on CelebA achieve a new state-of-the-art FID score of 1.85 for linear diffusion models”. We clarified this further everywhere in the paper and cited the [3], [4] works.
> >
> > Both [3], [4] are non-linear diffusion models. [3] trains a diffusion model on a latent space of a flow network. [4] is not a purely diffusion paper; it uses a discriminator and adversarial training. We want to clarify that the tricks introduced in [3], [4] are orthogonal to our formulation. We could also be training a blurring diffusion model on the latent space of a flow model or use a discriminator to enhance the quality of the generated samples as done in [4]. However, these experiments (even though interesting) are far beyond the scope of the paper.
> >
> > We want to emphasize that the main goal of our work is to convince the community that there is an alternative to vanilla Gaussian Diffusion and to provide the tools (new loss and sampler) to explore this direction. We believe this has more value than marginally pushing down the state-of-the-art FID score. In any case, the results of [4] on CelebA were reported after the ICLR deadline and [3] only slightly outperforms our method (reducing FID from 1.85 to 1.75) with tricks that are orthogonal to the choice of the corruption mechanism.
> >
> > **More details on estimating Wasserstein distances**
> >
> > It is true that estimating Wasserstein distances accurately is expensive. However, for finding a reasonable scheduling we do not need a very accurate estimation – we just need a sense of how far are the distributions at different scales to arrive at a smooth diffusion trajectory.
> >
> > For solving the optimal transport problem, we use the Sinkhorn Distances from the NeurIPS 2013 paper [Sinkhorn Distances: Lightspeed Computation of Optimal Transport](https://proceedings.neurips.cc/paper/2013/file/af21d0c97db2e27e13572cbf59eb343d-Paper.pdf) (that already experiments with image datasets). We use the implementation of this paper from the software package `ott-jax`. We experimented with both using the whole dataset and using slices. As expected, dataset slices lead to higher estimation errors, but we observe that the found scheduling doesn’t change much, i.e. the estimation error increases approximately uniformly for all the pairs for reasonably sized dataset slices. For the schedules found in the paper, we used slices of 10% of the dataset. Once we have an estimation of the pairwise distances from `ott-jax`, we find the scheduling using the parameters mentioned in Appendix C (Schedulings).
> >
> > We want to note that even though the scheduling with our method might be expensive, the cost is trivial compared to the training.
> >
> > We added this discussion and the experimental details in the paper (Appendix, Section D, Training Details). If the Reviewer has additional questions or suggestions on ways to estimate the Wasserstein distances, we would be happy to discuss more.
> >
> > **Suggestion for evaluating with different samplers**
> >
> > We thank the Reviewer for this valuable recommendation that improved our paper. We experimented with DDIM-type samplers, with Predictor-Corrector samplers and (further) with Probability Flow ODE samplers. The Results are given in Tables 2, 3, 4 of our revision. Here we summarize the general findings:
> >
> > * DDIM-type samplers significantly outperform samplers based on the Reverse SDE in the regime of low NFEs. FID improves from $30.31$ (Momentum Sampler) for 50 NFEs to $5.08$ with our new DDIM samplers. For higher NFEs, DDIM samplers perform worse than our momentum samplers.
> > * For deterministic samplers (DDIM and Probability Flow ODE), performance flattens for high NFEs. Our stochastic Momentum Sampler has a U-shaped performance plot, i.e. there is a sweet spot for intermediate sampling steps that gives the best performance. This is consistent with the observations from [Karras et. al.](https://arxiv.org/pdf/2206.00364.pdf) -- e.g. compare Figure 2 page 4 (deterministic sampling) with Figure 4, page 8 (stochastic sampling).
> > * Predictor-Corrector samplers have strong performance for both low and high NFEs. In certain regimes, Predictor-Corrector samplers outperform both the Predictor and the Corrector.

---

> > > ### Author Response · Authors · 2022-11-11
> > > **Response to Reviewer pNiX, Part 3: Theoretical Concerns and Clarifications**
> > >
> > > **"Theoretically, extending diffusion models from the isotropic corruption process to general non-isotropic corruption is quite straightforward based on original DSM work or recent."**
> > >
> > > We respectfully disagree with the Reviewer on this. In fact, the Denoising Score Matching objective, which provably learns the score function (see Vincent 2011 and Theorem 1 in our paper), does not work in practice for general corruption processes
> > > (due to its statistical inefficiency – see discussion in Section 3.1. of the paper, page 4).
> > > Objectives used in other (concurrent) works such as Cold Diffusion and Blurring Diffusion Models, predict heuristically directly the clean image with no score matching guarantees. Apart from lack of guarantees, the clean-prediction training loss performs worse; as we show, FID increases from $1.85$ to $5.91$. Our Soft Score Matching maintains the theoretical guarantees of optimizing for the score-function and also works better in practice.
> > >
> > > **"There is no theoretical evidence to justify why minimizing distance with the selected w2 distance is better."**
> > >
> > > This is a good point, we thank the Reviewer for raising it.
> > > Some of the widely used schedulings for the noise of vanilla Gaussian Diffusion models also don’t have theoretical guarantees. For example, the geometric progression used for the VE SDE in the papers NCSN, NCSNv2 and NCSNv3 also does not have any formal guarantees, at least as far as we are aware. The authors of NCSNv2 mention: “To give some theoretical guidance on finding good noise scales, we consider a simple case where the dataset contains only one data point”. Obviously, the datasets do not have one sample. Also, there is not an analysis on how this affects discretization errors. Despite the lack of theoretical guarantees, this scheduling has been widely used because of its strong experimental performance. Without assumptions on the real distribution, it might be impossible to design a scheduling that minimizes the discretization error – yet, simple heuristics like the one we propose might be useful. Experimentally, we show in the Appendix, Section F.3., that our scheduling scheme outperforms a natural baseline based on the VE SDE scheduling of NCSN++.
> > >
> > > **"[...] the SDE of the general corruption, which has a general linear shift term, can also be transformed into VESDE with a similar technique in [5] or exponential transformation introduced in [6]"**
> > >
> > > We think there might be a confusion here. Given a Forward Ito SDE, it is true that one can use many methods to solve the reverse SDE, including exponential integration as proposed in [6]. Also, [6] shows that when the Forward Ito SDE has a linear drift term, there is a whole family of reverse SDEs (parametrized by lambda) that samples from the correct distribution. All those statements are correct, but irrelevant to our problem.
> > >
> > > The reason is that our forward process is not an Ito SDE. Specifically, to get $x_t$ we apply the linear operator on $x_0$, not $x_{t-dt}$. There is no matrix that can be applied to a blurry and noisy $x_{t-dt}$ to get a (more) blurry and (more) noisy $x_t$. In other words, the considered corruption process is not Markovian and it cannot be described by a standard Ito SDE. The Momentum Sampler we introduce is derived by approximating the non-Markovian forward process with an appropriate Ito SDE – the main trick to do so is to replace $x_0$ in the forward process with the conditional expectation of $x_0$ given $x_t$. By doing so, we get a drift term that is independent of $x_0$ (it only depends on $x_t$) and hence we arrive at an Ito SDE. Given this new forward SDE, we can get the reverse SDE and use any of the tricks introduced in prior work (e.g. exponential integration introduced in [6]), to solve it. We hope this clarifies things and we thank the Reviewer for bringing it up. We updated the paper with this discussion, see also our response to Reviewer dDmh on a relevant question.
> > >
> > > Could the Reviewer elaborate more on the similar technique used in [5]? The authors are familiar with the work of Karras et. al, but we are not aware of such a derivation. We went through the paper again and we did not see any experiments (or new theory) for linear matrix drifts.
> > > The only slightly relevant idea we could find was some experiments with dataset augmentations. This is very far from what we are doing though. Karras et. al still use additive Gaussian noise at different scales to corrupt the image – augmentations are not (diffusion-) time dependent and they are used as a way to enrich the dataset and boost the FID score. This is mostly related to techniques previously used in GANs, e.g. see the [AmbientGAN](https://openreview.net/forum?id=Hy7fDog0b) paper and the [Training GANs with Limited Data](https://arxiv.org/abs/2006.06676) paper (again from Karras et. al). While the idea of using augmentations is worth exploring on its own, we do not see a direct connection between Soft Diffusion and this work.

---

> > > > ### Author Response · Authors · 2022-11-17
> > > > **Follow-up**
> > > >
> > > > We are happy to further discuss with the reviewer before the rebuttal stage is finished. Please let us know if you have more questions/comments.

---

> > > > > ### Comment · Reviewer_pNiX · 2022-12-04
> > > > > **Thanks for your response**
> > > > >
> > > > > Thanks for authors’ response and some discussions address my concern partially. The added DDIM experiments look promising. However, some discussions and conclusions still confuse me.
> > > > >
> > > > > 1. While eq-4 is a generalization of popular diffusion models, such as VPSDE and VESDE diffusion models[1] where $C_t$ is a scaled identity matrix, but it seems eq-13 can not reproduce the corresponding forward SDE discussed in [1].
> > > > > 2. And based on [2], for a large class $C_t$, it seems we can find corresponding SDEs such that the marginal distribution $q_t(x_t|x_0)$ is equivalent to the condition correction distribution stated in eq-4. I hope authors can discuss why we can not construct an Ito SDE such that the forward process shares the same marginal as eq-4, such as simple VPSDE and VPSDE. I understand it is difficult to find corresponding SDE if $C_t$ is not time continuous, but I fail to find assumptions for $C_t$ in the paper.
> > > > > 3. Can authors provide more justifications for eq-13, which is claimed that it is “close” to eq-4? The intuition built on the toy dataset is interesting. But for general distribution, can authors have any justifications for eq-13.
> > > > > 4. For scheduling based on w2 distance, it seems authors are estimating w2 distance with entropy-regularized stinkhorn distance. I think the scheduling is a more empirical choice instead of the ‘principled’ method stated in the paper.
> > > > > 5. Regarding Theorem 3.1, the only difference is a general matrix $C_t$ instead of the identity matrix introduced in original DSM[3]. The key ingredient [3, Appendix]~(similar to eq-26)  does not depends on the concretely form of $q_t(x_t|x_0)$. And proof of Theorem 3.1 is very similar to proof in [3, Appendix].
> > > > >
> > > > > [1] Song. Score-Based Generative Modeling through Stochastic Differential Equations
> > > > >
> > > > > [2] Zhang. Fast Sampling of Diffusion Models with Exponential Integrator
> > > > >
> > > > > [3] Vincent. A Connection Between Score Matching and Denoising Autoencoders

---

> > > > > > ### Author Response · Authors · 2022-12-09
> > > > > > **Further clarifications**
> > > > > >
> > > > > > We thank the Reviewer for reading our rebuttal. We are happy that the concerns of the Reviewers were addressed and we hope that the Reviewer will consider adjusting the score accordingly as they wrote in the review.
> > > > > >
> > > > > > We want to emphasize that the new questions raised by the Reviewer have been addressed in the revised version of our paper and in our responses to the rest of the Reviewers. For completeness, we explain briefly the main points here.
> > > > > >
> > > > > > **“it seems eq-13 can not reproduce the corresponding forward SDE discussed in [1]”**
> > > > > >
> > > > > > In Section 3.2 of the revised version we describe analytically the connection between our Momentum Sampler and the Forward SDE. We also justify theoretically why we can approximate both the conditional expectation and the score with our network, through Tweedie’s formula  (Lemma A.1.).
> > > > > >
> > > > > > We also want to emphasize that the experimental results clearly show that the Momentum Sampler works in practice – especially for higher number of steps, Momentum Sampling outperforms DDIM and predictor-corrector samplers.
> > > > > >
> > > > > > **“I hope authors can discuss why we can not construct an Ito SDE such that the forward process shares the same marginal as eq-4”**
> > > > > >
> > > > > > We do not think it is possible to describe the corruption process with an Ito SDE or at least we are not aware of how to do this for Gaussian blur + noise. Even in the case where the dataset has one sample, $x0$, the difference between $x_t$ and $x_{t - \Delta t}$ depends on $x0$ and hence the distribution of $x_t$ seems to depend on $x0$ and not only on $x_{t - \Delta t}$.
> > > > > > If the Reviewer has any suggestions on what is the underlying SDE, we would love to try it. The methods introduced in the paper “Zhang. Fast Sampling of Diffusion Models with Exponential Integrator” do not seem directly applicable – the authors analyze how to solve reverse SDEs, but for our corruption process we do not have a reverse SDE in the first place.
> > > > > >
> > > > > > **“Can authors provide more justifications for eq-13, which is claimed that it is “close” to eq-4? The intuition built on the toy dataset is interesting. But for general distribution, can authors have any justifications for eq-13.”**
> > > > > >
> > > > > > We point the Reviewer to the revisited Section 3.2 of our manuscript. We have significantly restructured this Section to explain much clearer the intuition behind our proposed SDE.
> > > > > >
> > > > > > Specifically, we introduced the Forward Ito SDE with the conditional expectation that is a Markovian Corruption Process. We then explained why this diffusion approximates reasonably well our non-markovian corruption process that we use during training. We explained the connection between our sampler and Tweedie's formula that allows us to approximate both the conditional expectation and the score with our neural network. We included Lemma A.1. to prove this connection (following the Tweedie's formula derivation).
> > > > > >
> > > > > > **“I think the scheduling is a more empirical choice instead of the ‘principled’ method stated in the paper.”**
> > > > > >
> > > > > > It is true that the proposed scheduling algorithm is heuristic. However, in the vast majority of the prior papers in the literature, the scheduling is also selected heuristically. In our paper, we proposed a method to choose the scheduling without manually tuning parameters that can work for any corruption. The proposed algorithm makes experimentation with different corruptions easy for any interested practitioner. Furthermore, there is recent evidence on the connection of optimal transport and diffusion models that we plan to cite in our next revision: https://openreview.net/forum?id=oPzICxVFqVM
> > > > > >
> > > > > > Finally, we want to point out that it is intuitive to have a scheduling that is dataset specific, as in our scheduling scheme. To the best of our knowledge, this idea has not been explored in the prior works.
> > > > > >
> > > > > >
> > > > > > **“And proof of Theorem 3.1 is very similar to proof in [3, Appendix].”**
> > > > > >
> > > > > > We have also clarified this in the revision of our paper and in our responses. Our Theorem identifies under which conditions the result from Vincent 2011 holds for other corruptions.

---

### Official Review · Reviewer_dDmh · 2022-11-04

**Confidence:** 4
**Correctness:** 3
**Technical Novelty And Significance:** 3
**Empirical Novelty And Significance:** 3
**Recommendation:** 5

**Clarity, Quality, Novelty And Reproducibility:**

Clarity (high): The clarify is generally clear, apart from some cases there are some inconsistencies with math notations (missing bold fonts for vector r.v.s, notation for Wiener process, $s_t$ only appears once).

Quality (high): there are some typos in the equations here and there, but generally it is okay.

Novelty (medium): technically this is not very surprising given existence of similar ideas, although there are some differences with prior work. The proposed SDE / ODE / samplers are not precisely tied to the original Fokker-Planck view of Gaussian diffusion SDE sampling, and can be viewed as an approximation of the original technique at best.

Originality (high): despite burgeoning similar ideas, the use of  is technically not explored before. It is a bit similar to the work on " Generative modelling with inverse heat dissipation", despite differences in training (denoising target is clean image instead of blurry image) and sampling (introduced momentum sampler).

**Strength And Weaknesses:**

*Strengths*

- Proposed a diffusion process combined with corruptions, such as blurring.
- A leap-frog like procedure is proposed for sampling, which addresses the issue of changing corruptions in the naive sampler.
- Competitive FID results on generative modeling.
- The authors are rather honest about the limitations of the method, as detailed in Appendix E.

*Weaknesses*

**Reverse SDE might not recover desired distribution**

The reverse SDE is based on an ad-hoc modification to the Gaussian diffusion SDE (specialized to the case where data distribution is delta), which makes it hard to justify why this would recover the desired data distribution (e.g., with Fokker-Planck). Wouldn't annealed Langevin dynamics / predictor-corrector be more principled?

**Experiments are limited**

All the experiments are done in CIFAR-10 and CelebA-64. While that is not anything to blame alone, it is difficult to generalize the conclusions to higher resolutions or even another dataset. One of the reasons is in Figure 3, it shows that the claimed SOTA FID is obtained by heavily tuning the number of NFEs, which is quite different from general findings that sample quality improves with more sample steps in standard diffusion models. Is this finding true for CIFAR-10 as well?

Also, some results seem to be missing.
- The Probability Flow ODE sampling with momentum is only listed for CIFAR-10 and not CelebA.
- CIFAR10 results should also have a table, with results on naive sampler as well.
- It does not seem unreasonable to train a model on VP-SDE / Elucidated DM given their superiority than VE.

The standard Gaussian diffusion models is a high bar to overcome, and it is unclear whether the experimental results would convince practitioners to switch to methods based on blurring. There does not seem to be advantages regarding architecture design, or sampling speed, or training speed. The most promising direction is to use these corruption models in inverse problems, as the authors suggested in Section 6, but there is no empirical evidence on this either.

**Novelty of Theorem 3.1**

Anything new about the proof of Theorem 3.1 that is not from Vincent (2011)? It appears all the tricks are the same, and $C_t$ does not even show up. The only change is $q_\sigma(\tilde{x} | x)$ (in Vincent's work that is a noise model) to $q(x_t | x)$ (that is the corrupted noise model), but the set of equations to prove the statements is basically the same.

**Inflated claims**

It is said that "Our method is superior in sampling time, for both CIFAR-10 and CelebA.". First, it is not mentioned the superior is against which method (likely VE-SDE, but that is probably the worst baseline for comparing sampling time). Second, the claim is not backed up for CIFAR-10 in the main paper (even in Figure 8, it is only shown for NFE within 100 and 170); only CelebA results are shown.


------

These are not weaknesses, but simple out of curiosity:

- I would recommend an ablation study on the effect on blurring degrees to the generative model, since you can interpolate between the blur kernel that is being used and doing no corruption at all. This could substantially improve the arguments on section 3.3, and indeed show its promise in generative modeling.

- The DDIM paper describes a series of solvers for the probability flow ODE and various SDEs (with different stochasticity in the sampling process). Is it possible to adopt that into this framework?

**Summary Of The Paper:**

The paper introduces a diffusion-based generative model trained on reversing a noisy and corruption process. Using the idea is that denoising score matching also works with general corruptions, score matching for corruption is proposed. To use it in a generative model, a momentum sampler is proposed on top of the naive sampler (which seems to have inferior performance). Experiments results on CIFAR-10 and CelebA show promising results in generative modeling, specifically on CelebA 64x64.

**Summary Of The Review:**

While learning diffusion models for blurring corruptions is interesting, the theorem is essentially rewriting the one in Vincent (2011), and the experiments are not comprehensive enough to demonstrate why one would favor this method than regular Gaussian diffusion. Even for a systematic study of soft diffusion alone, the work does not have a lot of CIFAR-10 results reported for CelebA, does not report NFE systematically (for many FID numbers except for the best ones, it is unclear how many NFE is used), does not have VP-SDE results, and does not explore ablation studies over the corruption process. Therefore, at the current stage, I am leaning towards rejection.

---

> ### Author Response · Authors · 2022-11-10
> **Response to Reviewer dDmh: Part 1, Summary of Revisions**
>
> We thank the Reviewer for their valuable comments. We are happy that the Reviewer finds our approach novel and our results competitive. The Reviewer raised some issues in two directions: i) significance of our theoretical results and ii) limited experimental evaluation.
>
> The feedback from the Reviewer genuinely helped us to improve our paper. We submitted to the system a revision of our paper that addresses the concerns raised by the Reviewer. In this response, we summarize the changes we made for visibility. Analytical responses for the additional Experimental Evaluation and Theoretical Concerns are given in the next comments.
>
> **Summary of changes**:
>
> * We show how we can use DDIM-type samplers in our framework, as suggested by the Reviewer. Our new DDIM samplers (presented in Section B2 of the Appendix, results in Table 3) significantly outperform our Momentum Sampler and our Probability Flow Momentum Sampler in a low number of function evaluations. FID improves from $30.31$ (Momentum Sampler) for 50 NFEs to $5.08$ with our new DDIM samplers. For higher NFEs, DDIM-type samplers perform worse than our samplers with momentum.
> * We experiment with Predictor-Corrector samplers (see Table 5), as proposed by the Reviewer. Predictor-Corrector samplers are strong in both the regimes of low and high NFEs, making it easier for practitioners to use our method without having to tune the optimal number of NFEs. For $\approx 100$ NFEs, Predictor-Corrector Samplers are better than both the Predictor and the Corrector.
> * We report missing results, as recommended by the Reviewer. Specifically, we add results for many different NFEs for the Probability Flow Momentum Sampler in CelebA (see Table 4). We also summarize our CIFAR-10 results in a Table (see Table 2) and we add Naive Sampler FID scores.
> * We show that for our deterministic samplers quality does not degrade much as the number of steps increases, hence tuning the NFEs is not required for strong performance. This is consistent with the results observed in "Elucidating the Design Space of Diffusion-Based Generative Models" – stochastic samplers might lose performance for many steps, but deterministic samplers don’t.
> * We explain more the theoretical underpinnings of the Momentum Sampler and when it  samples from the correct distribution.
> * We further clarify that our Theorem identifies under which conditions the result from Vincent 2011 holds for other corruptions.
> * We corrected several imprecisions in the manuscript that hopefully helps improve the understanding.
>
> We want to thank again the Reviewer again for their helpful comments. We detail these contributions and answer the Reviewer's questions
> in separate answers below. We would be glad to answer any further questions the Reviewer might have.

---

> > ### Author Response · Authors · 2022-11-10
> > **Response to Reviewer dDmh: Part 2, Theoretical Concerns and Clarifications**
> >
> > In this response, we address the theoretical concerns of the Reviewer and we clarify a few points about our method.
> >
> > **“Reverse SDE might not recover desired distribution”**
> >
> > We think there is some (understandable) confusion regarding the distribution our sampler is sampling from. This confusion stems primarily from the fact that we tried to derive our Momentum Sampler in an intuitive way in the paper, but that reasonably leads to questions regarding the sampling distribution and the approximations being made. We clarify below and we improved our presentation in our paper based on the Reviewer’s feedback.
> >
> > Consider the forward process:
> >
> > $dx_t = \dot C_t\mathbb E_{x_0 \sim p_0}[x_0|x_t] + g(t)dw$.
> >
> > This process is described by an Ito SDE, i.e. an SDE of the form:
> >
> > $dx_t = f(x_t, t)dt + g(t)dw$, where $f(x_t, t) =  \dot C_t\mathbb E_{x_0 \sim p_0}[x_0|x_t]$.
> >
> > The sampling process we write in the paper (that leads to the Momentum Sampler) is exactly the reverse of this forward process, where $\mathbb E_{x_0 \sim p_0}[x_0|x_t]$ is approximated by the neural network. Hence, the Fokker-Planck equations hold and we are guaranteed that (as long as the approximation of the conditional expectation and the approximation of the score function are accurate), we are sampling from the correct distribution.
> >
> > During training, we cannot use this process because we do not have the conditional expectation. Instead, we replace the conditional expectation of $x_0$ with $x_0$ itself. The mismatch between $x_0 $ and its conditional expectation leads to an additional approximation error in the learning of the score.
> > For small $t$, the conditional expectation will be close to $x_0$ and hence this approximation error is small. For large values of $t$, the distance between the conditional expectation and $x_0$ is bigger. However, we are multiplying with the corruption matrix $C_t$, which is removing information as $t$ grows. Therefore the distance of the corrupted conditional expectation and the corrupted $x_0$ is also expected to be small.
> >
> > Our training process learns the correct score for the corruption process applied directly to $x_0$. Our sampler, however, assumes that we have learned the score using the conditional expectation of $x_0$. We intuitively expect that these two are not far away (for all t) as we explained.
> >
> > We hope this is a helpful clarification. We added this discussion in the paper (Appendix Section E).
> >
> >
> > **“Novelty of Theorem 3.1”**
> >
> > The reviewer is right. We simply observe that the proof of Vincent (2011) holds for general  corruption processes, as long as some technical conditions are satisfied. We made this even more clear in the paper and stated this as an observation building on Vincent’s Theorem.
> >
> > The technical conditions are important for two reasons:
> >
> > 1. The key technical condition is that the corruption process assigns non-zero probability to any $x_t$ starting from any $x_0$. Blurring without noise (e.g. as done in Cold diffusion) does not satisfy this condition and hence does not learn the score. Yet, the authors demonstrated experimental success, which brings us to our next point.
> > 2. The Denoising Score Matching objective does not work in practice for general corruptions. This is because the model simply learns to denoise: there is no deblurring anywhere in the objective. Instead, Soft Score Matching, the objective we introduce, practically works while still maintaining the theoretical guarantees for learning the score function.
> >
> > Our point is that Denoising Score Matching doesn't work out of the box for general corruptions. Our key contribution is taking the loss in the filtered space to propose the Soft Score Matching objective.
> >
> > **“Is it possible to adopt DDIM into your framework?”**
> >
> > We thank the Reviewer for this valuable suggestion! By adopting DDIM samplers, we were able to outperform our samplers with Momentum in the low NFE regime. We added this to our paper, please refer to Appendix B.2 and F.3 (Table 2). Here, we explain briefly the idea.
> >
> > The idea is to use the network to predict $\hat x_0$ from $x_{t}$ and then use the forward model to move from $\hat x_0$ to $x_{t - \Delta t}$. This is same with the Naive Sampler we introduced in the main paper but the difference in DDIM is that part of the stochasticity is replaced by ``simulated" noise. The main trick to simulate the noise is to observe that once we know $\hat x_0$ and $x_t$, we can once again use the forward model to estimate the noise. We can use the same idea for Soft Diffusion sampling. Our DDIM-type sampler is given in Eq. (42) of the updated paper, also available [here](https://imgur.com/a/vwDKLWs).
> >
> > This Equation is very similar to Equation 12, page 5 in the DDIM paper. The parameter $k$ controls the amount of stochasticity, i.e. how much of the noise is simulated. For $k=0$, we get a deterministic sampler -- which is similar, but different to our Probability Flow Momentum Sampler. Happy to answer any additional questions.

---

> > > ### Author Response · Authors · 2022-11-10
> > > **Response to Reviewer dDmh: Part 3, Experimental Evaluation**
> > >
> > > In this response, we detail new Experimental Results we included in our revision and we answer related questions of the Reviewer. Once again, we want to thank the Reviewer for the valuable suggestions.
> > >
> > > **“Figure 3, it shows that the claimed SOTA FID is obtained by heavily tuning the number of NFEs, which is quite different from general findings that sample quality improves with more sample steps in standard diffusion models. Is this finding true for CIFAR-10 as well?”**
> > >
> > > It is true that the Momentum Sampler seems to have a U-shaped performance plot, i.e. there is a sweet spot in the number of function evaluations that gives the lowest FID score. We observed this in both CelebA and CIFAR-10.
> > >
> > > However, this is not unique to our sampler – the general belief that sample quality improves with the number of steps has been called into question by recent works. Specifically, Karras et. al. do an extensive evaluation of the role of NFEs for different samplers and they find that many stochastic samplers behave similarly to our Momentum Sampler. For example, we refer the Reviewer to Figure 4, page 8 of the paper Elucidating the Design Space of Diffusion-Based Generative Models that clearly shows that for many stochastic samplers performance slightly deteriorates after a certain threshold of function evaluations and the optimum is at some intermediate point.  We emphasize, however, that the deterioration we observed in CelebA is bigger compared to other stochastic samplers and therefore selecting the correct number of NFEs is important for our method. We made this clear in our paper.
> > >
> > > We also want to note that selecting the optimal number of function evaluations for reporting FID is not uncommon in the literature. It is done in both the landmark papers Elucidating the Design Space of Diffusion-Based Generative Models and Score-Based Generative Modeling through Stochastic Differential Equations. In our case, we did not do extensive finetuning – we selected the best FID among models evaluated with steps ranging from 200 to 1000 with step size 80 (10 experiments total). We reported this in the Appendix, we thank the Reviewer for bringing it up.
> > >
> > > This deterioration of performance that we observe for our Momentum Sampler is observed in Elucidating the Design Space of Diffusion-Based Generative Models only for the stochastic samplers. For example, Figure 2 page 4 of their paper shows the performance of deterministic samplers for which the phenomenon is to a very large extent mitigated.
> > >
> > > To test whether this is the case for our corruption as well, we experiment with two deterministic samplers:
> > > 1. Probability Flow Momentum Sampler (deterministic version of our Momentum Sampler, see page 6, Eq. (18) of our paper)
> > > 2. DDIM-type sampler, introduced in Section B.2. of the Appendix (see also previous answer)
> > >
> > > We report the performance as a function of the NFEs for CelebA at Tables 3, 4 for DDIM and Probability Flow Momentum Sampler respectively. We observe that both our deterministic samplers maintain much more the performance as the NFEs increases compared to the (stochastic) Momentum Sampler. Especially for DDIM, the performance is almost flat from 300 to 1000 steps. This is consistent with the behavior observed in Elucidating the Design Space of Diffusion-Based Generative Models. We hope this result clarifies the reasonable observation by the Reviewer. Even though for CelebA deterministic samplers seem to have worse performance (e.g. FID increases from $1.85$ to $2.14$), it might be helpful for practitioners to work with samplers that do not need careful tuning of the NFEs.
> > >
> > >
> > > **“The Probability Flow ODE sampling with momentum is only listed for CIFAR-10 and not CelebA.”**
> > >
> > >
> > > We thank the reviewer for catching this. We reported analytical FID results (as a function of NFEs) for Probabilty Flow ODE sampling in the Appendix, Table 4.
> > >
> > > **“CIFAR10 results should also have a table, with results on naive sampler as well.”**
> > > We thank the Reviewer for the suggestion. We included a Table (Table 2) in our revision, see also below:
> > >
> > > | Method                                          | FID         |
> > > |-------------------------------------------------|-------------|
> > > | Ours (VE SDE) Naive Sampler                     | 40.07       |
> > > | Ours (VE SDE) Momentum Sampler                  | 3.91        |
> > > | Ours (VE SDE) Probability Flow Momentum Sampler | 3.86        |
> > > | NCSN++ (VE SDE) Reverse SDE                     | 4.79        |
> > > | NCSN++ (VE SDE) Probability Flow                | 10.54       |
> > > | DDPM (VP SDE)                                   | 3.17        |
> > > | LSGM                   | **2.10** |
> > >
> > > Our best FID is competitive with similar samplers to similar methods, e.g. with NCSN++ trained with VE SDE and using the Reverse SDE sampler. It is also outperforming concurrent works for generalizing diffusion models, e.g. Cold Diffusion achieves FID $\approx 80$.
> > > Still, we are behind on CIFAR-10 compared to state-of-the-art such as LSGM.

---

> > > > ### Author Response · Authors · 2022-11-10
> > > > **Response to dDmh: Part 4, Experimental Evaluation (cont)**
> > > >
> > > > **DDIM-type samplers**
> > > >
> > > > We want to deeply thank the Reviewer for proposing the exploration of DDIM-type samplers. As we explained earlier, it is possible to adopt the DDIM sampling idea in our framework. By doing so, we get samplers that keep strong performance in very low number of function evaluations, maybe paving the way for wider adoption of diffusion models that revert more general corruptions. We report our DDIM results in Table 3 (for CelebA) and also below:
> > > >
> > > > | NFEs | FID  |
> > > > |------|------|
> > > > | 25   | 10.54 |
> > > > | 50   | 5.08 |
> > > > | 100  | 3.40 |
> > > > | 200  | 2.92 |
> > > > | 300  | 2.83 |
> > > > | 600  | 2.80 |
> > > > | 1000 | 2.79 |
> > > >
> > > > FID improves from $30.31$ (Momentum Sampler) for 50 NFEs to $5.08$ with our new DDIM samplers. For higher NFEs, DDIM-type samplers perform worse than our samplers with momentum.
> > > >
> > > > We also want to note that the performance of DDIM sampler does not degrade as we increase the number of steps, as we noted earlier.
> > > >
> > > > **“Wouldn't annealed Langevin dynamics / predictor-corrector be more principled?”**
> > > >
> > > > As explained in our response about the sampling distribution, our Momentum Sampler samples from the right distribution up to approximation errors in: i) learning the score-function and ii) estimating the conditional expectation.
> > > >
> > > > The idea of exploring Predictor-Corrector samplers is interesting though, we thank the Reviewer for suggesting it. We experimented with Predictor-Corrector samplers and we reported our Results in Table 5, see also below:
> > > >
> > > > | NFE  | FID   |
> > > > |------|-------|
> > > > | 25   | 10.56 |
> > > > | 50   | 5.21  |
> > > > | 100  | 3.21  |
> > > > | 200  | 2.86  |
> > > > | 300  | 2.71  |
> > > > | 600  | 2.73  |
> > > > | 1000 | 2.73  |
> > > >
> > > > We used a DDIM Predictor and a Probability Flow Momentum Sampler corrector. The Predictor-Corrector sampler maintains some of the benefits of both samplers. Namely, for low number of function evaluations it has better performance than the Probability Flow Momentum Sampler (benefit coming from the
> > > > DDIM sampler) and for a higher number of function evaluations performance is better than the DDIM sampler (benefit coming from the Probability Flow Momentum Sampler). There is a spot, at 100 NFEs, where the **Predictor-Corrector sampler is better than both the Predictor and the Corrector**.
> > > > It is possible that other pairs of Predictor-Corrector give further performance boost and we are happy to explore this more if the Reviewer finds it interesting.
> > > >
> > > >
> > > >
> > > >
> > > > **Other clarifications**:
> > > > * *“it is not mentioned which method you are superior to”*: We explicitly mention in the main paper (page 8, above Table 1): “Our method requires significantly fewer steps to achieve the same or better quality than NCSN++ (VE SDE)”. We compare with the VE SDE because our model was also trained with a VE SDE.
> > > > * *"It does not seem unreasonable to train a model on VP-SDE / Elucidated DM given their superiority than VE."* This is a fair point. We thank the Reviewer for the suggestion -- this would probably further improve the reported performance, at least on CIFAR-10. We started training a VP-SDE blurring model but due to limited access to computational resources we might not be able to have it until the rebuttal deadline. In any case, we want to emphasize that the main goal of our work is to convince the community that there is an alternative to vanilla Gaussian Diffusion and to provide the tools (new loss, sampler and scheduling technique) to explore this direction. We believe this has more value than marginally pushing down the state-of-the-art FID score -- which we already did for linear diffusion models on CelebA.
> > > > * *"the work does not explore ablation studies over the corruption process"*. We list Ablation Studies regarding the noise and the blur in Section F.1. of the Appendix. The Reviewer recommends ``an ablation study on the effect on blurring degrees to the generative model". We have an Ablation for a different scheduling for the blur, in Section F.1.2 of the Appendix. The change in the blurring scheduling leads to a dramatic deterioration of the FID score, that jumps from $1.85$ to $8.35$.
> > > >
> > > >
> > > > We thank again the Reviewer for the useful feedback and happy to answer more questions, if any.

---

> > > > > ### Comment · Reviewer_dDmh · 2022-11-14
> > > > > **Response to the rebuttal**
> > > > >
> > > > > Thank you for the effort in the rebuttal and the revised paper. The experiments on DDIM and PC samplers make the paper more comprehensive.
> > > > >
> > > > > I think most of the responses address my concerns, yet there are a few confusions.
> > > > >
> > > > > > Corruption process: I re-read section 3.3 and found it still to be confusing.
> > > > >
> > > > > - In this process does $D_1$ correspond to a Gaussian or the set of very blur images (but with little Gaussian noise)? It seems to be the latter from the experiment description, as noise is first added and then blur is added.
> > > > >
> > > > > - How is this path measured, since computing Wasserstein distance is quite non-trivial.
> > > > >
> > > > > > Consider the forward process: $dx_t = \dot C_t\mathbb E_{x_0 \sim p_0}[x_0|x_t] + g(t)dw$.
> > > > >
> > > > > This seems to generalize the one in the paper (for a fixed image $\alpha$), but I wonder if this forward process will indeed recover the desired marginal distribution over $x_t$ (which is blur to C_t, and add Gaussian noise)? I guess an underlying assumption is that $E_{x_0 \sim p_0}[x_0|x_t]$ gives you a expression over the score by Tweedie's formula, but here the corruption is wrt. a blur process instead of adding Gaussian noise. While you can derive Tweedie's formula for any member of the exponential family, does that give you the same equations as you would get in the Gaussian case? Or, because blur is linear, so it is still a Gaussian case and everything still works the way it is? I feel that this (and the momentum sampler equations) would need some further explanation, and it is not as trivial as it seems.

---

> > > > > > ### Author Response · Authors · 2022-11-16
> > > > > > **Rebuttal Follow-up: Scheduling and Sampling**
> > > > > >
> > > > > > We thank the Reviewer for engaging in the rebuttal. The review process has strengthened our paper. We are glad that most of the concerns are now addressed. We further clarify the remaining issues in our latest revision (Nov. 15, 2022) and in the comments below.
> > > > > >
> > > > > > **“In this process does $D_1$ correspond to a Gaussian or the set of very blur images (but with little Gaussian noise)? It seems to be the latter from the experiment description, as noise is first added and then blur is added.”**
> > > > > >
> > > > > > The distribution $\mathcal D_1$ is the distribution of very blurry images with additive Gaussian noise. At the limit, each blurry image becomes a constant image having a single color (i.e., the average color of the image). Hence, to sample from this final distribution, we first have to sample a single color (from the distribution of average colors in the dataset), generate a constant image having that color, and then add little noise to the constant image. The distribution of average colors for all the considered datasets is very simple and we model it with a Gaussian distribution. Specifically, we fit a three-dimensional gaussian distribution (one dimension for each color channel, diagonal covariance) to the average color of the dataset. To begin the inference process, we sample one color from this distribution, we create an image where every pixel has this color and then we add Gaussian noise. We hope this clarifies what the final distribution is. We added this discussion and further details in Section D of the Appendix.
> > > > > >
> > > > > >
> > > > > > **“How is this path measured, since computing Wasserstein distance is quite non-trivial.”**
> > > > > >
> > > > > > It is true that computing Wasserstein distances accurately is not trivial. For finding a reasonable scheduling we do not necessarily need a very accurate estimation – we just need a sense of how far are the distributions at different scales to get a smooth diffusion trajectory.
> > > > > >
> > > > > > For solving the optimal transport problem, we use the Sinkhorn Distances from the paper Sinkhorn Distances: Lightspeed Computation of Optimal Transport (NeurIPS 2013 paper, that experiments with image datasets). We use the implementation of the package `ott-jax`. We experimented with both using the whole dataset and using slices. As expected, dataset slices lead to higher estimation errors, but we observe that the found scheduling doesn’t change much, i.e. the estimation error increases approximately uniformly for all the pairs for reasonably sized dataset slices. For the schedules found in the paper, we used slices of 10% of the dataset. Once we have an estimate of the pairwise distances, we find the scheduling as explained in Appendix C.
> > > > > > We want to note that even though the scheduling with our method might be expensive, the cost is insignificant compared to the training.
> > > > > >
> > > > > > **Connection of Momentum Sampler to Tweedie’s formula**
> > > > > >
> > > > > > The Reviewer is absolutely right, there is a connection between our sampler and Tweedie’s formula. Following the reviewer's suggestion, we use Tweedie’s formula to connect the conditional expectation to the score function. Hence, the only unknown is the score function which is approximated by our neural network. We include the proof at Lemma A.1. for completeness (but it tightly follows the Tweedie’s formula derivation).
> > > > > >
> > > > > > Using the feedback of the Reviewer, we restructured the Momentum Sampling Section to illustrate this connection and clarify the underlying assumptions behind the derivation of our sampler. In what follows, we summarize the key points:
> > > > > >
> > > > > > * We introduce directly the Forward Ito SDE with the conditional expectation. This corresponds to a Markovian Corruption process.
> > > > > > * We then look at the toy setting where the dataset has one sample and we show that the marginals of the training corruption process and the Markovian SDE are the same for this simple case.
> > > > > > * We clarify that in the general setting this is not necessarily the case –by approximating the corruption process with a Markovian process we introduce some approximation error. This error seems negligible in practice, given the experimental success of our Momentum Sampler. We explain intuitively why this is the case. Namely, the conditional expectation is close to the clean image itself for low corruption (small approximation error). For high corruption levels the distances are contracted by our corruption matrix $C_t$.
> > > > > > * We write the reverse SDE (and the Probability Flow ODE) of the Ito SDE that has the conditional expectation.
> > > > > > * We prove (and use) that there is a connection between the conditional expectation and the score through Tweedie’s formula. The proof is given in Lemma A.1. in the Appendix. Using this Lemma, the only unknown is the score function, which is approximated by our network.
> > > > > >
> > > > > > We want to thank the Reviewer for the useful comments and suggestions. The presentation of our Momentum Sampler is much more clear now. Also, the additional sampling experiments helped us make our paper more comprehensive, as the Reviewer noted.

---

> > > > > > > ### Comment · Reviewer_dDmh · 2022-11-19
> > > > > > > **Proof of Lemma A.1**
> > > > > > >
> > > > > > > I apologize in advance, but I am not able to read too closely into the proof or give comments yet due to the CVPR supplementary deadline. Unfortunately, it seems that this is the deadline for draft revision as well.
> > > > > > >
> > > > > > > But I will take a closer look at the statement and proofs later (from a skim, it looks similar to how you would prove Tweedie's formula, which is quite reasonable).

---

> > > > > > > > ### Author Response · Authors · 2022-12-09
> > > > > > > > **Follow-up?**
> > > > > > > >
> > > > > > > > We wanted to follow-up and see if the Reviewer had the time to go over our rebuttal.
> > > > > > > > Since we put a lot of effort into significantly improving our paper, following the Reviewer's feedback, we would like to kindly ask the Reviewer to let us know if their concerns have been addressed and consider updating the recommendation accordingly.
> > > > > > > >
> > > > > > > > We remain in the disposal of the Reviewers and the AC if further questions arise.

---

### Author Response · Authors · 2022-11-19
**General Response**

We thank the Reviewers for their constructive feedback. The review process helped us significantly strengthen our paper.
During the rebuttal period, we extensively replied to issues raised by the Reviewers, we performed many additional experiments to strengthen the experimental evaluation of our work and we clarified confusions. We would really appreciate it if the Reviewers read our responses and our revised paper and adjust their evaluations if their concerns got resolved. Below, we summarize the key changes:

* We show how we can use DDIM-type samplers in our framework. Our new DDIM samplers (presented in Section B2 of the Appendix, results in Table 3) significantly outperform our Momentum Sampler and our Probability Flow Momentum Sampler in the low NFEs regime. For higher NFEs, DDIM-type samplers perform worse than our samplers with momentum.
* We experiment with Predictor-Corrector samplers (see Table 5). Predictor-Corrector samplers are strong in both the regimes of low and high NFEs, making it easier for practitioners to use our method without having to tune the optimal number of NFEs. In certain regimes, PC samplers outperform both the Predictor and the Corrector.
* We report missing results. Specifically, we add results for many different NFEs for the Probability Flow Momentum Sampler in CelebA (see Table 4). We also summarize our CIFAR-10 results in a Table (see Table 2) and we add Naive Sampler FID scores.
* We add visual comparisons for fixed NFEs to the previous state-of-the-art model (DDPM++), as suggested. We show (visually, on top of the quantitative results) that our models outperform prior state-of-the-art models (with the same type of samplers) for constrained computational budgets, even for CIFAR10.
* We show that for our deterministic samplers quality does not degrade much as the number of steps increases, hence tuning the NFEs is not required for strong performance.
* We significantly restructured our section on Momentum Sampling to explain better the theoretical underpinning behind the algorithm. We
introduce the Forward Ito SDE with the conditional expectation that is a Markovian Corruption Process. We then explained why this diffusion approximates reasonably well our non-markovian corruption process that we use during training. We explain the connection between our sampler and Tweedie's formula that allows us to approximate both the conditional expectation and the score with our neural network. We included Lemma A.1. to prove this connection (following the Tweedie's formula derivation).
* We further clarify that our Theorem identifies under which conditions the result from Vincent 2011 holds for other corruptions.
* We clarify that, by the time of the submission, our method achieves state-of-the-art on CelebA among linear diffusion models. Among all the available models, to the best of our knowledge, our method ranks 2nd.
* We explain in detail the difference of our work with other works that consider linear drifts. In short, our corruption process is not Markovian and hence it cannot be described by an Ito SDE. The novelty of the Momentum Sampler is approximating the corruption process with a suitable Ito SDE (that can later be solved with any of the methods in the literature).
* We further explain the differences of Soft Score Matching with previously proposed objectives in the literature (Cold Diffusion, DSM).
* We provide further details on the computation of Wasserstein distances that are needed for the proposed unsupervised scheduling algorithm.
* We provide a lot more implementation details to ensure that our results are reproducible. Specifically, we give pseudo-code on how to generate the blurring kernels and the masking arrays and we provide our raw parameters through anonymized links.

We thank again the Reviewers for their constructive feedback and we hope that the issues they raised are now resolved. We are happy to answer additional questions, if any.

---

### Decision · Program_Chairs · 2023-01-20

**Decision:**

Reject

**Justification For Why Not Higher Score:**

The proposed framework contains potential flaws. None of the reviewers feel comfortable recommending acceptance.

**Justification For Why Not Lower Score:**

N/A

**Metareview: Summary, Strengths And Weaknesses:**

The paper proposes a diffusion process that can include various corruption types, such as blur. It uses a leap-frog-like sampling procedure and achieves competitive FID scores on benchmark datasets.

The paper receives three reviews. All the reviewers rate the paper below the bar. Several weakness includes potential flaws in the proposed framework, limited experiment results, and over-claiming. The authors, based on the reviewers' suggestions, include a large set of new results. The submission is substantially different from the original one. However, none of the reviewers update their ratings despite the authors' consistent push. The AC consults with several reviewers separately, confirming that they have read the rebuttal and updated the paper. However, it appears they still have doubts about the acceptance of the paper.

After reading the paper, reviews, and rebuttal, the AC feels the reviewers' concerns are justified. The paper isn't convincing enough in its current form and requires additional work to make the cut.

**Summary Of Ac-Reviewer Meeting:**

N/A